# Human spatiotemporal pattern learning as probabilistic program synthesis

**Tracey E. Mills**
MIT
temills@mit.edu

**Joshua B. Tenenbaum**
MIT
jbt@mit.edu

**Samuel J. Cheyette**
MIT
cheyette@mit.edu

## Abstract

People are adept at learning a wide variety of structured patterns from small amounts of data, presenting a conundrum from the standpoint of the bias-variance tradeoff: what kinds of representations and algorithms support the joint flexibility and data-paucity of human learning? One possibility is that people "learn by programming": inducing probabilistic models to fit observed data. Here, we experimentally test human learning in the domain of structured 2-dimensional patterns, using a task in which participants repeatedly predicted where a dot would move based on its previous trajectory. We evaluate human performance against standard parametric and non-parametric time-series models, as well as two Bayesian program synthesis models whose hypotheses vary in their degree of structure: a compositional Gaussian Process model and a structured "Language of Thought" (LoT) model. We find that signatures of human pattern learning are best explained by the LoT model, supporting the idea that the flexibility and data-efficiency of human structure learning can be understood as probabilistic inference over an expressive space of programs.

## 1 Introduction

Humans, uniquely among animals, are exceptional learners of structure: we learn languages, develop and use systems of mathematics, make music, follow recipes, and play board games. It is very unlikely that we have evolved specialized modules for these specific abilities, save language, given their breadth and the fact that many were invented quite recently. Instead, people seem to have powerful inference mechanisms that enable learning and representing structured theories in many different domains. Yet the flexibility of human learning presents a challenge to models of intelligence based, for instance, on training large, data-intensive networks. While there are now machine learning systems that successfully mimic or even surpass certain human abilities, humans have general inductive abilities that exceed modern AI systems and, importantly, often require much less data.

Models of human structure learning historically fall under two broad approaches: symbolic approaches, based on grammars and logic, and statistical approaches, such as neural networks and Bayesian inference. Symbolic approaches have focused on capturing the productivity, systematicity, and compositionality of human thought [1]; however, they have often been implemented in ways that are insufficiently flexible to handle ambiguity and uncertainty which are also central features of cognition. In contrast, statistical approaches have rarely enforced this same degree of structure in learned representations, in order to achieve greater flexibility and robustness to ambiguity and uncertainty. Recently, the idea of modeling human learning as *probabilistic program induction* — in which probabilistic inference operates over structured, symbolic programs — has emerged as a compelling synthesis of these two approaches [2].

Program induction models have shown success in explaining patterns in human learning in a number of domains, including logical concepts, handwritten characters, counting, kinship relations, sequences,

37th Conference on Neural Information Processing Systems (NeurIPS 2023).

list functions, and question asking, among others [e.g. 3–10]. For instance, Piantadosi et al. [6] used a program learning model containing primitives of objects, sets, set operations, and recursion to explain how children learn number words, capturing their progression from understanding the meaning of "one," "two," "three," and "four," to being "Cardinality Principle" knowers: understanding the relationship between the count list and cardinalities in general. Though this and other work supports the empirical validity of modeling human learning as probabilistic inference over programs, the evidence has largely been indirect, e.g. comparing relative learning rates of different concepts in people and models. Only in very constrained domains, such as learning concepts corresponding to Boolean or first-order logic programs, have there been systematic attempts to assess the specific inductive biases employed by human learners or to measure directly the extent to which people are making probabilistic inferences over candidate programs [3, 4].

There has been some work assessing the degree of programmatic structure present in people's inductive biases when learning in more complex or naturalistic domains. For example, past work has found that people have *compositional* inductive biases when learning 1D functions or time-series. Schulz et al. [11] built on Gaussian Process (GP) models of 1D function learning [12, 13] to show that human learners prefer compositional hypotheses over non-compositional alternatives matched for low-level visual properties. Specifically, Schulz et al. investigated models of human function learning as Bayesian inference over a grammar of GP kernels which specify the features of high probability functions (e.g. periodic, smoothly varying, linearly increasing). These models are capable of explaining complex patterns in data by combining kernel primitives to form compound kernels (e.g., a periodic deviation on top of a smoothly increasing trend) [14–16], while also maintaining a bias for simple functions by enforcing a prior for simple kernel structures. Though these compositional GP models offer one formalization of the joint flexibility and structure characteristic of human learning within this relatively unconstrained hypothesis space, they do not attempt to capture the full range of algorithmic structure that people can learn.

Other research has investigated human learning of more general program structures in the setting of 2D geometry and geometric sequence learning. Amalric et al. [8] found that both adults and children are strongly biased toward sequences described by programs that involve operations with symmetry, regular rotations, and repetitions in time. Relatedly, Sablé-Meyer et al. [17] argued that people represent geometric shapes in terms of high-level program control structures such as repetition, concatenation, and embedding, and found that participants' complexity ratings and memory of shapes was better predicted by a shape's minimum description length in a grammar that includes these concepts as primitives than in one which does not. However, these studies either did not directly model inference or did not test program induction against alternative accounts of learning.

Thus, although a strong basis has begun to form for the "learning as programming" account, there is still relatively little work that aims to thoroughly compare human inferences to those of both probabilistic program induction models and other plausible inductive models. That is our goal here. We experimentally test human predictions on a spatiotemporal pattern learning task that features a wide range of algorithmically interesting geometric structures, including variants of spirals, zig-zags, polygons, grids, staircases and other repeating and nested combinations of these motifs. In each trial of the task, participants repeatedly made predictions about where a dot would move next within a 2D region, based on previously observed points in a discrete sequence. While this task is quite simple, it requires participants to navigate a rich and theoretically unconstrained hypothesis space on the basis of very little data, with the potential to give insight into both the inductive biases and representations of uncertainty in people's hypotheses.

Our analyses compare the probability distributions of participants' predictions to those of multiple varieties of Bayesian program induction models as well as several competing accounts. Models tested include Bayesian polynomial regression, a Gaussian Process model that learns both a kernel and parameters, and two Bayesian program synthesis models with grammars that afford different degrees of flexibility and structure: a highly flexible Gaussian Process model that jointly learns kernel structure and parameters, and a structured "Language of Thought" (LoT) model with more traditional programming primitives such as loops, motor commands, and arithmetic operators. We also describe preliminary explorations with an LSTM-based neural network account.

We find that the two Bayesian program learning models provide the best accounts of both the mean performance of human learners and the distributions of their predictions at successive points of the observed sequences. The fine-grained signatures of human pattern learning, including both patterns

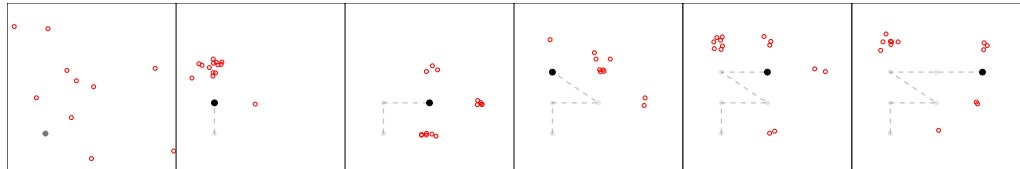

Figure 1: The first six timepoints of a sequence, shown to illustrate the task. The most recently revealed point in the sequence is shown as a black dot, previous points are shown in lighter gray, and human predictions are shown as red circles.

of successful and incorrect predictions, are best explained by the Bayesian LoT model, supporting the idea that the flexibility and data-efficiency of human structure learning can be understood as probabilistic inference over an expressive space of explicitly structured programs.

## 2 Experiment

### 2.1 Method

We recruited 50 adult participants from the online platform Prolific who each completed 20 trials in which they learned a unique sequence of 2D points. Participants were asked to predict successively the location of each point in the sequence conditioned on having seen the locations of all previous points. Each sequence contained 21 points including the initial point, so that participants made 20 predictions, with up to four attempts per prediction per trial. After completing each trial, participants responded to two questions with a sliding scale: "How confident are you that you learned the pattern?", from "not confident (0)" to 'very confident (100)" and "How complicated was the pattern?", from "not complicated (0)" to "very complicated (100)." See Figure 1 for an illustration of the prediction task and the distribution of participants' predictions on each of the first six timepoints of one sequence. The set of all sequences is shown in full in Figure S3 in the Supplementary Materials.

At the beginning of each trial, a single point was shown on the screen within a display region with aspect ratio 4:3 and width in pixels $w$ which was maximized based on a participant's screen dimensions. Participants were instructed to click where they predicted the next point would appear. Participants were allowed to click within a gray circular region which was centered at the previous point with radius $0.21 * w$ pixels, and which excluded the area within $0.020 * w$ pixels of the previous point (see Figure S2 in the Supplementary Materials). If participants guessed within an "acceptance distance" ($0.026 * w$ pixels) of the next point in the sequence, their prediction would be considered correct and the next point would be revealed. Participants were given up to four incorrect attempts before the next point was revealed. Previous points in the sequence remained visible, but turned lighter according to how far back in the sequence they appeared. Participants were paid \$7.25 for completion of the study, with a median completion time of 46 minutes.

### 2.2 Results

The full set of participants' predictions on all 50 sequences is visible in Figure 2, with early points in cooler colors (blue/purple) and later points in warmer colors (orange/yellow). Participants generally succeeded at the task, making correct predictions on their first attempt for 11.97 of the 20 predictions per sequence on average, though the number of correct first-attempt predictions varies considerably across sequences (see Figure 3). To more precisely quantify the extent to which participants successfully learned each sequence, we compute the relative absolute error of a participant's prediction at each timepoint as the absolute distance between their first prediction attempt [1] and the true point, divided by the distance between the true point and the previous true point. We excluded the 10% of participants with the greatest mean relative absolute error across timepoints and sequences from our analyses.

Under this metric, participants successfully learned almost every pattern, with mean relative absolute error across participants and timepoints below 1 (the mean relative absolute error that would result if the location of each prediction attempt was simply the previous true point) for 49/50 sequences.

---

[1]Although participants were given up to four prediction attempts at each timepoint, we only include the first attempt in our analyses. The use of multiple attempts in the experiment primarily served as motivation for participants to remain engaged in the task.

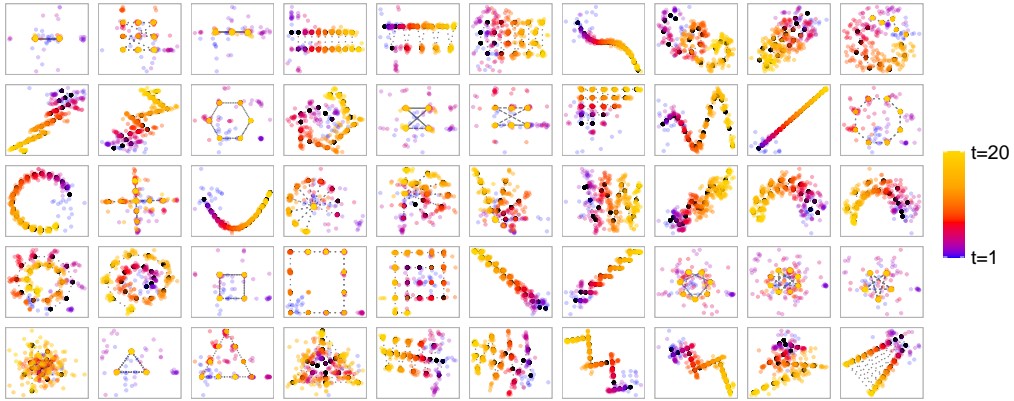

Figure 2: Human predictions for each sequence across all timepoints, with predictions for earlier timepoints shown in cooler colors (blue/purple) and those for later timepoints in warmer colors (orange/yellow).

Learning varied noticeably by sequence, with mean relative error across participants and timepoints ranging from 0.11 to 1.04. Participants became more accurate as time went on within each sequence, with an average relative error across sequences of 0.94 at the first timepoint, 0.21 at the final timepoint, and 0.43 across timepoints. Figure 3 shows the distribution of the number of correct predictions (within $0.026 * w$ pixels) participants made on their first attempt across all sequences. There are some sequences, such as a noiseless line, that participants got correct on their first attempt for all but the first or second prediction. On the other end of the spectrum, there are some sequences which participants struggled to learn even after 15 data points (e.g. sequences 49 and 50, as shown in Figure S3 in the Supplementary Materials). Overall, mean relative error had a strong positive relationship with participants' complexity ratings ($r = 0.77$, $p < .001$), and a strong negative relationship with confidence ratings ($r = -0.87$, $p < .001$).

## 3 Models

We evaluate several models on our task: first, we evaluate two models which have been used to explain human function learning in past work [12, 18–20], which fit parameters of either a fixed structure or a structure selected from a highly constrained set: a Bayesian polynomial regression model, and a non-compositional Gaussian Process (GP-NC) model where a single kernel and parameters are inferred. We also evaluate two *structure-learning* models: a Gaussian Process model where the compositional kernel structure and parameters are jointly inferred (GP-SL), and a Bayesian LoT program-learning model over motor and geometry primitives.

Each model was implemented in Gen [21] using Sequential Monte Carlo (SMC) with Markov Chain Monte Carlo (MCMC) rejuvenation steps. Specifically, all models run SMC with 20 particles which, after an additional observation becomes available, are resampled if their effective sample size falls below 10. There are then 100,000 rejuvenation steps on the inferred hypothesis (including noise parameters) for each particle. The rejuvenation steps use a generalization of Metropolis Hastings called Involutive MCMC that allows for custom-built kernels in the reversible jump MCMC framework. After rejuvenation, each resulting particle specifies a predicted location and noise estimate over this prediction for the next point in the sequence. Model predictions at each timepoint are marginalized over particle predictions, weighted by their posterior probability.

### 3.1 Fixed-structure models

**Bayesian Ridge regression**

Ridge regression is a linear modeling technique that uses $\ell_2$ regularization to constrain fit model coefficients — a natural choice for a problem involving sparse data and highly correlated variables, which both apply to this task. Here, we use a Bayesian variant of Ridge regression, which finds

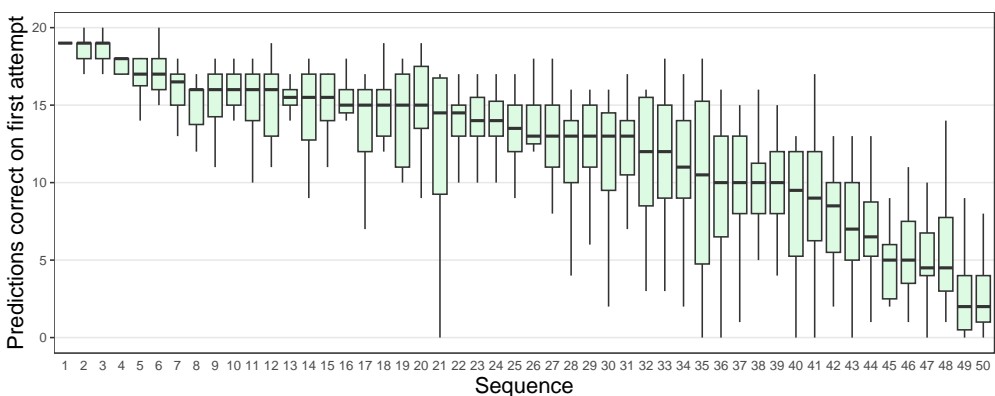

Figure 3: Boxplots for each sequence of the number of timepoints at which participants correctly predicted the location of the next point on their first attempt, with outliers removed. Sequence numbers correspond to those in Figure S3 in the Supplementary Materials.

estimates for regression weights β assuming a spherical Gaussian prior $p(\beta \mid \lambda) = \mathcal{N}(\beta \mid 0, \lambda^{-1}\mathbb{I}_p)$, where λ is a prior on the precision. The goal is to learn a mapping from previously observed $x$, $y$ coordinates and the time $t$ that they were observed, to the next $x$ value and next $y$ value in the sequence — i.e., the $x$ and $y$ values at time $t+1$. This is done via two regressions: one to predict subsequent $x$ values and one to predict subsequent $y$ values. The mean predicted values have the form:

$$\mu_x = \beta_{x,0} + \beta_{x,1} \cdot t + \beta_{x,2} \cdot x + \beta_{x,3} \cdot y$$
$$\mu_y = \beta_{y,0} + \beta_{y,1} \cdot t + \beta_{y,2} \cdot x + \beta_{y,3} \cdot y$$

The likelihood is then the product of $\mathcal{N}(x \mid \mu_x, \sigma_x)$ and $\mathcal{N}(y \mid \mu_y, \sigma_y)$.

In the main text, we show results only for the linear model with weight coefficients on $x$, $y$, and $t$, but in the Supplementary Materials we also show results for the Bayesian Ridge regression model using 2 and 3 degree features (i.e., $x$, $y$, $t$ and their two or three degree interactions, such as $x^2$, $x \cdot y$, $x \cdot t$, etc.).

**Non-Compositional GP model**

Inspired by the model developed in [12], we fit a Gaussian Process model that jointly infers both the kernel and its parameters, but does not allow compositional kernel structures (e.g. $K_1(\cdot) \cdot K_2(\cdot)$). We use four base kernels each with at least one numeric parameter: **Constant** (with a value parameter), **Linear** (with an intercept parameter), **Radial Basis Function** (with a scale parameter), and **Periodic** (with scale and period parameters). Each kernel specifies a prior over functions and each is useful for modeling particular types of data; for instance, the Periodic kernel is often used for modeling periodically varying data, such as weather patterns. We assume that $x$ and $y$ observations are generated by independent kernels, which each take the previous $x$ value, previous $y$ value, or the current $t$ value as an input; one kernel outputs a prediction for $x$ and the other for $y$ at time $t+1$. Each base kernel additionally has a categorical parameter that specifies which dimension ($x$, $y$, or $t$) to take as input, allowing for inference about which input dimension is most predictive. We used uniform priors on the kernel type and on noise. The likelihood is determined by the probability of drawing the vector of true $x$ and $y$ values at each timepoint from a multivariate normal distribution with a mean vector of zeros and covariance matrix as determined by the noise parameter and kernel function for that dimension.

### 3.2 Program synthesis models

Both the GP-SL and LoT models have probabilistic context free grammars (PCFGs) which define a space of hypothetical program trees. During MCMC rejuvenation, Metropolis Hastings proposal moves manipulate the structure of inferred program trees as well as their parameters (see [22]).

**GP Structure-Learning model**

The GP-SL model, adapted from the one presented in [15] to handle multidimensional output, uses the same base kernels as the GP-NC model but additionally allows for kernel operators that *compose* the

base kernels. Specifically, the grammar additionally includes three composite kernels: **Sum**, **Product**, and **Changepoint** (with a location parameter which specifies an input point at which to smoothly transitions between kernels). During inference, Metropolis Hastings proposals over program tree structures include tree regeneration moves, which involve the replacement of random subtrees with trees generated from the prior distribution. As with the GP-NC model, the GP-SL model infers independent *x* and *y* output kernels and noise parameters, and the likelihood is determined by the probability of drawing the vector of true *x* and *y* values at each timepoint from a multivariate normal distribution with a mean vector of zeros and covariance matrix as determined by the noise parameter and kernel function for that dimension. In this case the prior is determined by the PCFG from which the *x* and *y* kernels are drawn in addition to the uniform prior on noise.

### LoT Model

The LoT grammar draws inspiration from [17], which defined a simple Logo-like drawing language to model human representations of geometric shapes. This language includes a mixture of control operations (e.g. loops), motor commands (e.g. turn), and numeric expressions, which in combination efficiently generates myriad canonical and more complex geometric shapes. In this work, Sablé-Meyer et al. found that the complexity of MDL programs in this grammar predicted people's ease of encoding and recalling the shapes generated by those programs. Our LoT grammar is organized similarly, but has some added complexity to allow for greater flexibility in the range of sequences it can generate. In particular, our grammar includes access to inner-state variables such as the current position or timepoint, as well as a wider range of motor commands and algebraic operators, making it highly expressive. Programs drawn from this grammar define algorithms for generating unboundedly-long sequences of *x* and *y* positions over time.

| Operations (*o*) | | Values (*v*) | | |
|---|---|---|---|---|
| (Controls) | (Actions) | (State variables) | (Numbers) | (Expressions) |
| Repeat(*o*, *v*) | Move() | $\theta$ (current angle) | $\mathbb{N}$ (naturals) | Plus(*v*, *v*) |
| Continue(*o*) | Stay() | *s* (current speed) | $\mathbb{R}$ (reals) | Minus(*v*, *v*) |
| Concat(*o*, *o*) | Turn(*v*) | *x* (current x-position) | | Times(*v*, *v*) |
| Subprogram(*o*) | Accelerate(*v*) | *y* (current y-position) | | Divide(*v*, *v*) |
| | ChangeX(*v*) | *t* (current time) | | Mod(*v*, *v*) |
| | ChangeY(*v*) | *n* (function calls) | | Sin(*v*) |
| | SetX(*v*) | | | |
| | SetY(*v*) | | | |

Table 1: LoT model primitives.

Table 1 shows the full set of primitives in the grammar. *Operations* define the structure of the program, and *values* are numeric parameters to operations. Operations include *control nodes* which act over other operations: **Repeat**, which when evaluated repeats a given operation a certain number of times; **Continue**, which repeats a given operation indefinitely (when being evaluated, each program generated by our model is wrapped in an outer **Continue** call which repeatedly executes the program until it generates a sequence of the specified length); **Concat**, which sequentially executes two operations; and **Subprogram**, which executes an operation and then restores the previous internal program state. Operations also include *action nodes*: **Move** and **Stay** determine the next location in the sequence by setting the position at the next timepoint. **Move** changes the current position based on the current speed and angle, and increments the current time, while **Stay** increments the current time without changing the current position. Other action nodes simply update the internal state at the current timepoint: **Turn**, **Accelerate**, **ChangeX**, and **ChangeY** update the current angle, speed, and *x* and *y* positions respectively by a given value. **SetX** and **SetY** simply set the current *x* and *y* positions to a given value.

Value nodes include numbers (**Naturals** and **Reals**) as well as the current values of internal state variables: the angle ($\boldsymbol{\theta}$), where a value of 1 corresponds to 90 degrees, speed ($\boldsymbol{s}$), *x* and *y* positions ($\boldsymbol{x}$, $\boldsymbol{y}$), timepoint ($\boldsymbol{t}$), and number of times the program has been executed by the outer **Continue** call ($\boldsymbol{n}$). Values can also be composed in expressions with **Plus**, **Minus**, **Times**, **Divide**, **Mod**, and **Sin** nodes.

This model can compactly represent many simple structures, such as geometrical shapes. For instance, the simplest program generating a square is **Repeat(Concat(Move(), Turn(1)), 4)**. However, by allowing sub-programs, recursive calls, and unbounded concatenation, the model

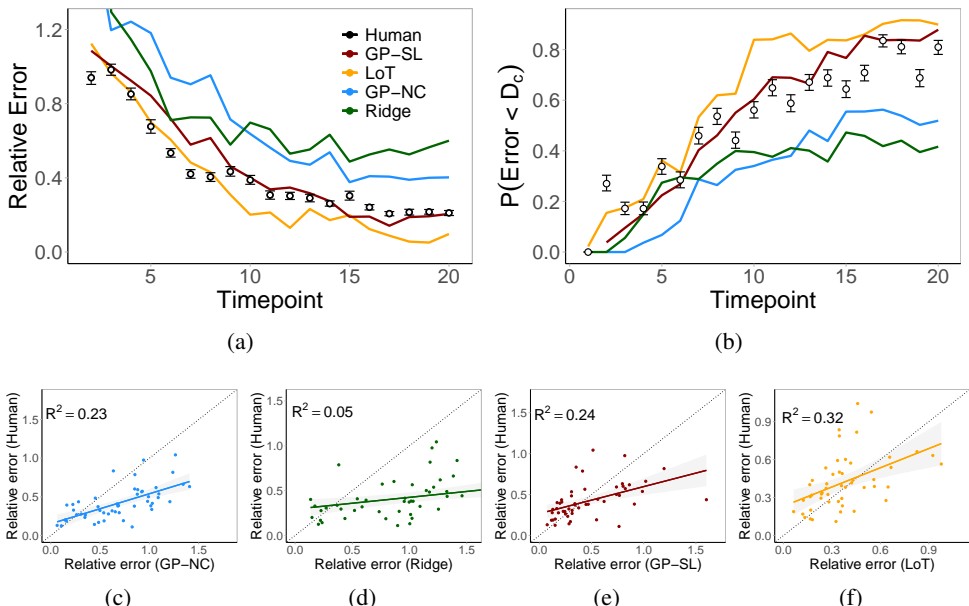

Figure 4: Panels (a-b) show learning curves averaged across sequences as a function of time. Panel (a) shows relative error over time and (b) shows the proportion of errors lower than the "acceptance distance" $D_c$ used to determine correctness in the human experiment. Note that error is defined as the distance between the prediction and the true point relative to the distance between the true point and the previous point in the sequence. Errorbars on human data points are bootstrapped 95% confidence intervals. Panels (c-f) show the correlation between human error (y-axis) and model error (x-axis) for all sequences tested. Panel (c) shows the GP-NC model; (d) shows the Bayesian Ridge regression; (e) shows the GP-SL model; and (f) shows the LoT model.

can also represent much more complex sequences at the cost of program length. The expressiveness of this grammar affords flexibility in the representations learned by the LoT model, but also presents a challenge for inference as it defines an extremely large, discrete search space. To address this challenge we defined several custom proposal moves on the program tree structure to more effectively explore the search space: in addition to subtree regeneration moves, at each rejuvenation step we include moves which swap single nodes with nodes of a different type, remove nodes and replace them with one of their children, and insert subtrees between parent and child nodes [23]. Involutive MCMC allows custom kernels as long as you can specify an invertible function that maps from any proposal back to the initial hypothesis, and we used Gen's built-in correctness checks to ensure there were no bugs in involution [21].

During inference, the posterior probability of a particle is computed based on its prior probability and likelihood. The prior is determined by the PCFG from which the program tree is generated, gamma distributions over the noise parameters and hyperparameters, and distributions over initial values of the state variables $\theta$ and $s$ (see the Supplementary Materials for a more detailed description). The likelihood is decomposed by the output dimension ($x$ or $y$), and for a single dimension is computed based on the probability of drawing, at each timepoint, the true location from a normal distribution centered around the predicted location at that timepoint, with the variance determined by the inferred noise parameters.

We analyze the performance of each model on our task with reference to human performance. We propose that the representational flexibility of the two program synthesis models is key to learning in this deceptively simple domain. We also aim to elucidate the nature and degree of structure in participants' inductive biases based on the explanatory power of each model.

### 3.3 Model comparison

The top row of Figure 4 (panels a-b) show aggregate learning curves for the models and participants over time, averaged across all sequences. Panel (a) shows relative error over time and panel (b) shows

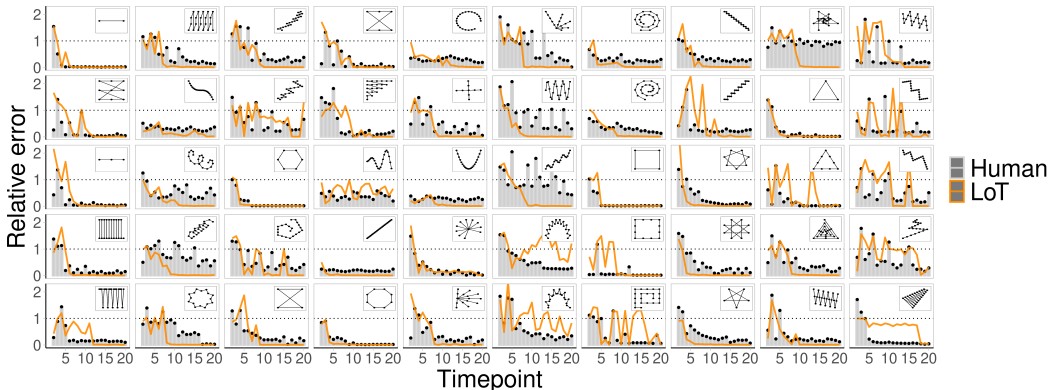

Figure 5: Absolute error relative to the distance from the previous point across all sequences tested for LoT model (orange) and human data (gray bars). The sequence corresponding to the data is shown in the top-right of each panel.

the proportion of absolute errors less than the "acceptance distance" $D_c$ used to determine correctness in the human experiment. The GP-NC model starts off with very high error and plateaus in learning at about timepoint 10, with roughly no subsequent improvement beyond. The Bayesian Ridge regression model performs similarly, though with slightly lower relative error after about timepoint 10. On the other hand, both the LoT model and the GP-SL models continuously improve, perhaps plateauing toward the final timepoint, and are both roughly in line with aggregate human performance. Panels (c-f) compare the average relative error of humans (y-axis) and each model (x-axis) across sequences, to determine the extent to which people and the model had difficulty with the same sequences. In order of lowest to highest correlation were the Ridge model ($r = 0.23$; $p < 0.001$), GP-NC model ($r = 0.48$; $p < 0.001$), GP-SL model ($r = 0.50$; $p < 0.001$) and the LoT model ($r = 0.56$; $p < 0.001$).

To quantitatively compare the fit of the models to the human data, we first fit parameters to capture two types of additional noise in the human data that the models were not intended to capture. Specifically, we assumed that participants' clicks had added motor noise and variability stemming from perceptual imprecision, and we also assumed that on some proportion of trials, participants may not have been attending or making a serious effort. We thus optimized the likelihood of participants' responses for each model assuming they were drawn from a mixture model of the form,

$$\hat{x}_{t+1} \sim (1 - \theta_l) \cdot \mathcal{N}\left(\mu_{x,\,t+1},\ \sigma^2_{x,\,t+1} + \sigma^2_m\right) + \theta_l \cdot \mathcal{N}\left(x_t,\ \sigma^2_l\right) \tag{1}$$

$$\hat{y}_{t+1} \sim (1 - \theta_l) \cdot \mathcal{N}\left(\mu_{y,\,t+1},\ \sigma^2_{y,\,t+1} + \sigma^2_m\right) + \theta_l \cdot \mathcal{N}\left(y_t,\ \sigma^2_l\right), \tag{2}$$

where the free parameters are the motor noise term $\sigma_m$, a lapse rate $\theta_l$, and noise in responses on lapse trials $\sigma_l$. The models' predictions for the $x$ and $y$ coordinates at time $t + 1$ are given by $\mu_{x,\,t+1}$ and $\mu_{y,\,t+1}$, respectively, and the predicted standard deviations are given by $\sigma_{x,\,t+1}$ and $\sigma_{y,\,t+1}$. Because the Bayesian Ridge regression and GP models required two training points to make predictions, we compared each model's fit to the data from after the second time-point for each sequence. We used R's built-in 'optim' function for fitting [24].

| Model type | $\sigma_m$ | $\theta_l$ | $\sigma_l$ | LL | $\Delta$AIC |
|---|---|---|---|---|---|
| GP-NC | 0.05 | 0.32 | 2.28 | -31592 | 19514 |
| Ridge | 0.01 | 0.20 | 1.96 | -30187 | 16704 |
| GP-SL | 0.16 | 0.20 | 1.60 | -28416 | 13162 |
| LoT | 0.01 | 0.24 | 1.74 | -21835 | 0 |

Table 2: Model parameters, Log Likelihood, and $\Delta$AIC from the best-fitting model.

Table 2 shows the fit parameters for each model, the resulting log-likelihood of the human data, and the $\Delta$AIC from the best-fitting model. The worst-fitting model was the GP-NC model, followed by the Bayesian Ridge regression. The GP-SL model and the LoT model both learned many more of the sequences, but the LoT model provided a significantly better fit to the human data, with a lower AIC score by 13162 (where 10 is often considered a "significant" difference).

Figure 5 shows the relative error (y-axis) at each timepoint (x-axis) made by people (gray bars/points) and the LoT model (orange line) for all sequences tested. The learning rate of the LoT model aligns quite closely with that of people for many of the sequences tested, which is more impressive considering that this figure shows model predictions absent any parameters fit to the data. Analogous figures for the other models are shown in the Supplementary Materials. Figure 6 provides a few illustrative examples of individual timepoints in various sequences of participants' (red points) and the particles of the LoT model's (blue points) predictions. These were chosen to highlight cases where the LoT model captures patterns of successes and errors in the human data, but also show cases where the predictions of the LoT model differ from people (rightmost column). The LoT model reproduces certain biases evident in the human data, such as biases toward repeating previous changes in angle or position, that result in multimodal clusters of uncertainty. The LoT model's uncertainty clusters do not always align with people's, however, and in general the model is more strongly biased toward simple functions that are a poor fit to the data early on in a sequence (e.g. learning just the function `Move()`, as in the failure case at the top of the rightmost column). All the models' predictions for each sequence across all timepoints are provided in the Supplementary Materials.

## 4   Discussion

Our results have shown that people can learn a variety of algorithmically-structured 2D sequences from a small number of data points. Yet, while participants learned nearly all the sequences by the final timepoint, many of the sequences we tested were not learned by a Gaussian Process model which selects from several different kernels. It is therefore an interesting question, given that participants were never exposed to these exact sequences, what makes people so adept at learning seemingly arbitrary patterns that non-parametric function approximators struggled to fit. One possible answer is that people have been exposed to similar stimuli in their past, and thus can solve the task with visual pattern recognition. Yet the space of potential sequences we could have picked that might look "familiar" to many people is vast, and thus their success seems unlikely to be attributable to a lifetime of data about, e.g., the space of all possible zig-zagging sequences. Even with large amounts of relevant data, however, amortizing useful inductive biases for this task may be difficult (though see [25, 26]). As a preliminary test, we trained an LSTM to predict thousands of sequences (sampled from the LoT model), which it did relatively well, but nonetheless generalized poorly to the sequences used in our experiment (see the Supplementary Materials).

We propose instead that people's success at learning algorithmic patterns comes from powerful inference mechanisms that are broadly applicable across many domains, ranging from those that are useful for physical reasoning (e.g. perceiving occluders [27]), to those that support more abstract human inventions (e.g., geometry [28]). Namely, people are able to infer "mental programs" to represent latent structure and uncertainty in data. This idea is supported by the success of the two Bayesian program synthesis models at learning the vast majority of the sequences we tested at a rate similar to people, in aggregate. In addition to aligning with the speed of human learning, the LoT model in particular — which explicitly represents sequences as an algorithm specifying how a point's trajectory should change in angle and speed as a function of time — recapitulates many human-like biases as well. For instance, the LoT model is biased toward repetition and "simple" angles (e.g. 90-degree or 45 degree). Gaussian process kernels, on the other hand, have no biases (or very weak ones) toward geometric regularities and symmetries.

Whether other animals, including even our close primate ancestors, represent structure in visual patterns in the same way as people is a particularly interesting unsettled question [29, 30]. Recent evidence suggests, in fact, that non-human primates do not represent shapes in the same way that humans do [31]. The authors of that study used an oddball paradigm to determine how western-educated adults, kindergartners, Himba adults without formal schooling, and baboons distinguished outlier geometric shapes in a set. They found that while all human subjects showed a geometric regularity effect, such that more regular geometric shapes were more easily detected as outliers, this was not the case for the baboons. The authors interpreted this as evidence that humans engage a "logical, recursive mode of representation akin to a programming language" absent in other animals.

Though our model comparison results primarily shed light on the nature and degree of structure in people's hypotheses, a key aspect of each of our models is that they reason probabilistically over these hypotheses. We see signatures of this probabilistic reasoning in human responses as well. Qualitatively, the patterns of multimodal uncertainty evident in the human data are reflective of a *distribution* over distinct hypotheses. For example, Figure 1 illustrates a case in which different

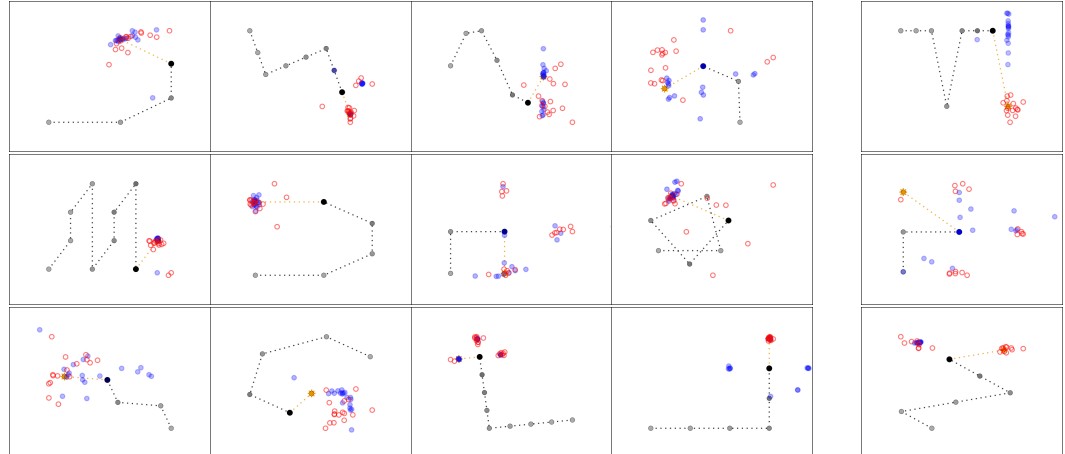

Figure 6: A set of examples of human (red) and LoT model (blue) predictions (SMC particles) chosen to highlight interesting cases. Examples in the rightmost column illustrate cases where the model makes predictions that are both wrong and also diverge from people. The transparency of the LoT model points is set by the posterior probability of the particle. The previous point in the sequence is shown as a black dot and earlier points in the sequence are shown in lighter gray. The true next point (i.e., ground-truth location) is shown as a gold star.

participants' responses are consistent with alternative hypotheses involving repetition of observed sequence substructures and features of geometric regularity. This is most evident at the third timepoint, with the three previous points forming a right angle. Participants' predictions are clustered around three points: left and up, straight, or right and down. That individuals seem to pursue distinct, well specified hypotheses, while in aggregate human response patterns indicate uncertainty in inference, is well captured by accounts of human learning and decision-making which hold that human predictions are based on one or few samples from a probability distribution over hypotheses [32].

In this work, our analyses have been at the computational level, testing whether the problem being solved by participants might be specified as Bayesian inference over a particular class of programmatic hypotheses. However, other work suggests that our models might additionally capture key algorithmic aspects of human inference. In particular, Thaker et al. [33] found that SMC offers a plausible algorithmic account of how people track multiple hypotheses over time, with limited-size particle filters explaining order effects in learning. Our models also employ MCMC rejuvenation moves, which Dasgupta et al. [34] have identified as an algorithmic explanation for dependencies between people's sequentially generated hypotheses [34]. Even more specifically, Franken et al. [35] present evidence that people iteratively modify hypotheses through processes akin to the tree-surgery proposal moves utilized by the LoT model during MCMC rejuvenation. Though we adopted these specific techniques because of their usefulness in navigating large, discrete hypothesis spaces, as opposed to their plausibility as algorithmic models of human behavior, this convergence speaks to the fruitfulness of jointly developing efficient modeling techniques and algorithmic accounts of human inference.

Finally, it is worth noting that people's inductive biases may have been shaped during the experiment by the specific stimuli we tested, which all had some geometric regularity and were free of noise (although the log likelihood of participant predictions under the LoT model actually decreases very slightly as they progress through trials ($r = -0.016$, $p < .05$)). The ability to construct and adaptively update domain-specific hypothesis spaces is a remarkable feat of human learning which the field of program synthesis has only begun to make progress toward incorporating [2, 9, 16, 36, 37]. Despite the expressiveness of the GP-SL and LoT models, they are unable to capture the range of hypotheses participants might theoretically consider (as an extreme example, we might imagine sequences which begin to spell out words). Computationally formalizing people's ability to flexibly construct and revise hypothesis spaces during online probabilistic inference is an exciting direction for future work in both cognitive science and machine learning, and the domain of 2D sequence learning provides a rich test bed for understanding the sensitivity of people's hypothesis spaces to variations in underlying data generating processes.

## Acknowledgements

This work was supported by the AFOSR Grant No. FA9550-19-1-0269, the DARPA Machine Common Sense Program, the ONR Science of AI Program, and Siegel Family Endowment. We would like to thank Mathias Sablé-Meyer, Stan Dehaene, Vikash Mansinghka, and especially Feras Saad for helpful and instructive conversations.

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
