# Supplementary Materials

Experiment and model source code, videos showing human and model predictions for all sequences, and examples of model hypotheses are available at `https://github.com/temills/spatiotemporal_patterns`.

## 1 Supplementary Methods

### Human Experiment

Below, we provide illustrations to show the experimental method in greater detail than provided in the main text. Figure S1 shows the exact instructions shown to participants. Figure S2 illustrates the experimental method with screenshots of a trial for the first three timepoints. A participant's task was to indicate where they thought the next point in a sequence would appear by clicking anywhere within the gray region, which surrounded the previous point's location. All the sequences used in the task are shown in Figure S3.

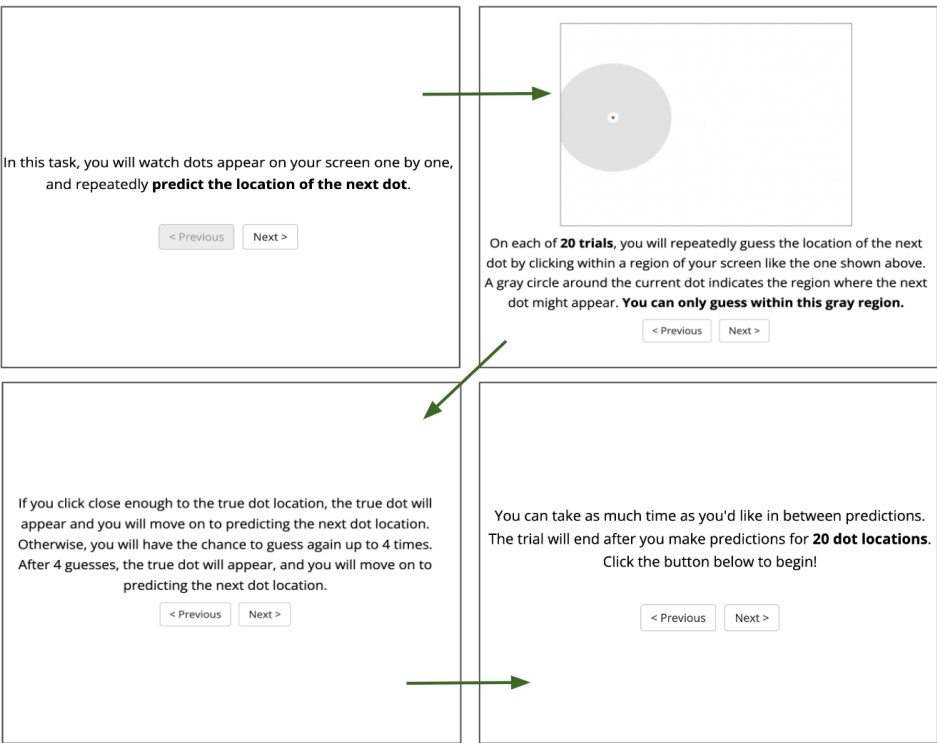

Figure S1: Instructions shown to human participants before taking part in the task, by page (order indicated by green arrows).

37th Conference on Neural Information Processing Systems (NeurIPS 2023).

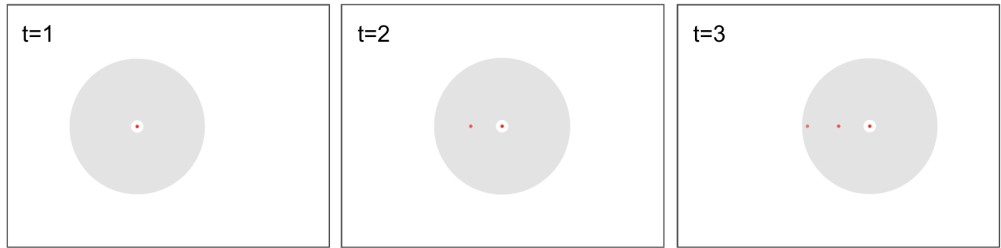

Figure S2: Selected screenshots from a single trial of the experiment, annotated by timepoint. At each timepoint, participants guessed the location of the next point in the sequence by clicking within the shaded region surrounding the previous point.

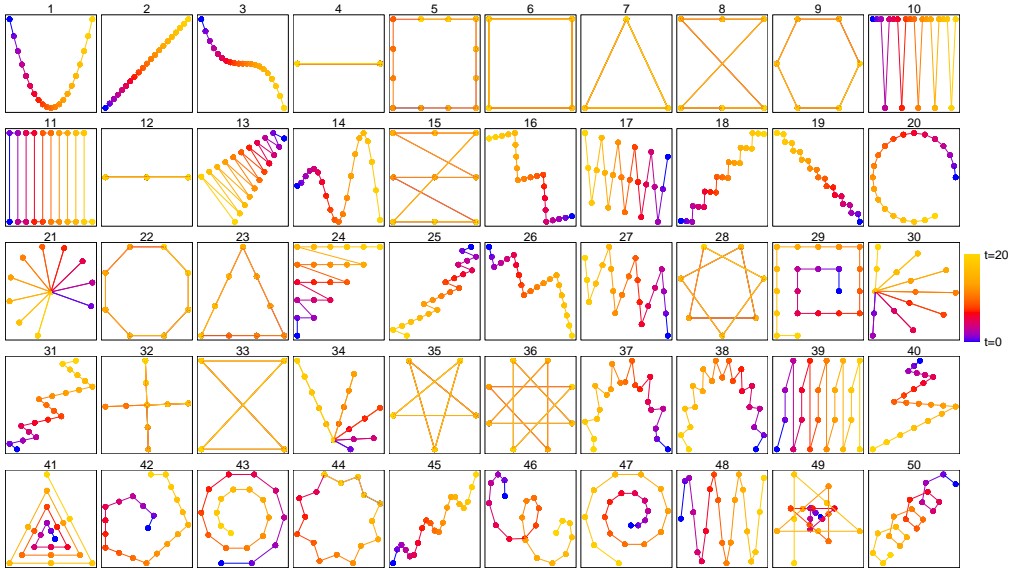

Figure S3: All sequence stimuli used in the experiment. Earlier points are shown in cooler colors (blue) and later points in warmer colors (yellow).

## Models

All models were implemented in Julia [1] using the probabilistic programming library Gen [2] for inference. In Gen, the assumptions of the problem-domain are encoded in a probabilistic generative function, which defines a data-generating process involving random choices. In each of our models, the generative function includes random choices determining the synthesized programs, parameters, and data (i.e. a sequence of points), and thus specifies a joint probability distribution over these. During inference (implemented with Sequential Monte Carlo with Markov Chain Monte Carlo rejuvenation as described in the main text) we then approximately sample from this distribution, conditioned on data. Each model had 20 particles for SMC and used 100,000 MCMC rejuvenation steps after each time point. Each model re-sampled noise, hyperpriors, and real-valued parameters using Gaussian drift, with the program synthesis models also re-sampling program trees as described in the main text.

### Fixed-structure models

**Bayesian Ridge Regression Model**

In classical Ridge regression, the goal is to minimize the objective function $\|y - \beta X\|_2^2 + \alpha \cdot \|\beta\|_2^2$, where $y$ is some observed data, $\beta$ is a set of weights, $X$ are features (i.e., $x$, $y$, and $t$), and $\alpha$ is a regularization term setting the penalty on large $\beta$ coefficients. Bayesian Ridge regression allows the hyperparameter on the weight regularization term, and the amount of uncertainty in the weights, to be inferred from the data. This then takes the form $p(y \mid \lambda, \alpha, X) = \mathcal{N}(y \mid \beta \cdot X, \alpha)$. $\alpha$ and $\lambda$ represent the precision of the noise and the precision of the weights, respectively. These parameters are assumed to be drawn from a Gamma distribution, with shape and rate (inverse scale) hyperparameters set to 1,

and fit jointly with the weight coefficients to the data. All $x$, $y$, and $t$ values were standardized across the entire sequence.

We also ran polynomial Ridge models incorporating higher order terms and interactions between the features to capture any non-linear relationships in the data. The hyperparameters in these polynomial models are learned exactly as in the linear case, with the addition of more terms in the $\beta$ vector corresponding to the additional features in the polynomial models. The 2nd-degree polynomial Bayesian Ridge regression model incorporates squared terms of the features and their pairwise interactions, so $x^2$, $y^2$, $t^2$, $x \cdot y$, $x \cdot t$, and $y \cdot t$ are included in addition to the first degree terms. These additional terms help the model capture relationships that might not be evident in the linear model. The 3rd-degree model allows cubic terms for each feature, and interactions up to the third degree, i.e. the terms $x^3$, $y^3$, $t^3$, $x^2 \cdot y$, $x \cdot y^2$, $x^2 \cdot t$, $x \cdot t^2$, $y^2 \cdot t$, $y \cdot t^2$ and $x \cdot y \cdot t$ are included in addition to the first and second degree polynomial terms. In each case, regressions predicting subsequent $x$ and $y$ values from previous $x$, $y$, and $t$ values were done separately for $x$ and $y$.

### Non-Compositional GP Model

Our GP-NC model had four base kernels: Constant, Linear, Radial Basis Function, and Periodic. Below, we provide the mathematical formulations for each of these kernels and detail their typical use cases.

### Constant Kernel

$$K(\mathbf{x}, \mathbf{x}') = c^2, \tag{1}$$

where $c$ is a constant.

### Linear Kernel

The Linear kernel is a simple kernel that allows the GP model to predict a linearly varying function. This kernel is useful when the underlying relationship between the inputs and outputs is assumed to be linear, with constant variance. Given input vectors $\mathbf{x}$ and $\mathbf{x}'$, the Linear kernel takes the form,

$$K(\mathbf{x}, \mathbf{x}') = \sigma^2 + \mathbf{x}^T \mathbf{x}', \tag{2}$$

where $\sigma^2$ is a constant offset.

### Radial Basis Function Kernel

The Radial Basis Function (RBF) kernel is one of the most commonly used kernels in GP regression. The RBF kernel characterizes the similarity between input vectors $\mathbf{x}$ and $\mathbf{x}'$ based on the Euclidean distance between them. Mathematically, this takes the form,

$$K(\mathbf{x}, \mathbf{x}') = \sigma^2 \exp\left(-\frac{(\mathbf{x} - \mathbf{x}')^T (\mathbf{x} - \mathbf{x}')}{2l^2}\right), \tag{3}$$

where $\sigma^2$ represents the overall variance of the data and $l$ is the length-scale hyperparameter controlling how quickly the similarity between two data points decays as their distance increases.

### Periodic Kernel

Lastly, the Periodic kernel is typically used to model periodic functions. It is a product of an exponential and a sine-squared term. This kernel is particularly useful for data that exhibits cyclical behavior, like weather patterns or seasonal trends. The Periodic kernel is defined as:

$$K(\mathbf{x}, \mathbf{x}') = \sigma^2 \exp\left(-\frac{2\sin^2(\pi \frac{\mathbf{x} - \mathbf{x}'}{p})}{l^2}\right), \tag{4}$$

where $p$ is the period of the function and $l$ is the length-scale hyperparameter.

The GP-NC model assumed that there were two independent kernels for the $x$ and $y$ dimensions. There was a uniform prior on each kernel and a uniform prior on an input dimension ($x$, $y$, or $t$) that it operated over. There was also a uniform prior on noise added to the covariance matrix. To compute the likelihood of the data given a kernel and its parameters, the sampled kernels are used to compute covariance matrices between two dimensions, e.g. $K_{X,Y}$. The diagonals of these matrices determine variance at a particular point, and the sampled noise value is added to the diagonal. Output values are then sampled from multivariate normal distributions parameterized by $\vec{\mu} = 0$ and the covariance matrices.

**Program synthesis models**

**LoT Model**

> **function** GENERATIVEFUNCTION($T$)
>> $P \sim PCFG$
>> $\theta \sim \text{Uniform}(-4, 4)$
>> $s \sim \text{Exp}(0.5)$
>> $concat\_move \sim \text{Bernoulli}(0.5)$
>> $\vec{\mu}_x, \vec{\mu}_y = \text{RUNPROGRAM}(P, \theta, s, T, concat\_move)$
>> $\alpha_x \sim \text{Gamma}(10, 10)$
>> $\beta_x \sim \text{Gamma}(1, 1)$
>> $\eta_x \sim \text{Gamma}(\alpha_x, \beta_x)$
>> $\alpha_y \sim \text{Gamma}(10, 10)$
>> $\beta_y \sim \text{Gamma}(1, 1)$
>> $\eta_y \sim \text{Gamma}(\alpha_y, \beta_y)$
>> **for** $t \leftarrow 1$ to $T$ **do**
>>> $x_t \sim \mathcal{N}(\mu_{x,t}, (1/\eta_x)^{0.5})$
>>> $y_t \sim \mathcal{N}(\mu_{y,t}, (1/\eta_y)^{0.5})$
>> **end for**
> **end function**

Here, the generative function specifies a joint probability distribution over the program, initial values of the angle and speed internal state variables, noise parameters, and the $x$ and $y$ location of each point in the sequence. The input to the function is $T$, the number of points in the sequence so far. In the generative function, the initial program $P$ is sampled from the PCFG as described in the main text. We also sample initial values for the internal state variables $\theta$ and $s$ which determine the starting speed and heading for $P$. We also sample a *concat_move* variable as a parameter to RunProgram, which returns the sequence of $x$ and $y$ locations ($\vec{\mu}_x$ and $\vec{\mu}_y$) specified by $P$ and these parameters.

In RunProgram, $P$ is evaluated as follows. As described in the main text, the internal state variables angle ($\boldsymbol{\theta}$), where a value of 1 corresponds to 90 degrees, speed ($\boldsymbol{s}$), $x$ and $y$ positions ($\boldsymbol{x}$, $\boldsymbol{y}$), timepoint ($\boldsymbol{t}$), and number of program runs ($\boldsymbol{n}$), are continuously tracked and updated. At the beginning of each timepoint, $\boldsymbol{x}$ and $\boldsymbol{y}$ are set to the true x and y position of the point observed at the previous timepoint, so that $P$ determines the movement from the previous true point to the next point at each timepoint. Specifically, each call in $P$ either specifies the control structure of the program, updates the internal state, or specifies the next point in the sequence and increments the current timepoint $\boldsymbol{t}$ with a **Stay** or **Move** call.

When **Stay** is called, the current $\boldsymbol{x}$ and $\boldsymbol{y}$ values are added to $\vec{\mu}_x$ and $\vec{\mu}_y$ respectively. When **Move** is called, $\boldsymbol{x}$ and $\boldsymbol{y}$ are updated according to the current speed and angle of movement, and these updated values are added to $\vec{\mu}_x$ and $\vec{\mu}_y$ respectively.

$P$ is wrapped in an outer **Continue** call which repeatedly runs the program, and increments the internal state variable $\boldsymbol{n}$, until $\vec{\mu}_x$ and $\vec{\mu}_y$ are of length $T$. Programs without a call to **Move** or **Stay** will not produce output, and so $\vec{\mu}_x$ and $\vec{\mu}_x$ are set to $[x_0]_T$ and $[y_0]_T$ respectively. This idiosyncrasy means that otherwise reasonable programs which simply lack a **Move** or **Stay** call will be evaluated as extremely unlikely, presenting a challenge for search. To combat this, we sample an additional *concat_move* variable which, if set to true, concatenates an additional **Move** call to $P$. This allows for more efficient exploration of the hypothesis space.

After running $P$, point locations are then sampled based on the output $\vec{\mu}_x$ and $\vec{\mu}_y$ and the sampled noise hyperparameters ($\alpha_x, \beta_x, \alpha_y, \beta_y$) and parameters ($\eta_x, \eta_y$).

**GP Structure-Learning model**

**function** GENERATIVEFUNCTION(input)
    $n = $ LENGTH($input$)
    $kernel\_x \sim$ PCFG
    $noise\_x \sim$ Uniform$(0.001, 0.1)$
    $cov\_mat\_x = $ COMPUTECOVARIANCEMATRIX$(kernel\_x, \text{input}) + \mathbb{I}_n * noise\_x$
    $kernel\_y \sim$ PCFG
    $noise\_y \sim$ Uniform$(0.001, 0.1)$
    $cov\_mat\_y = $ COMPUTECOVARIANCEMATRIX$(kernel\_y, \text{input}) + \mathbb{I}_n * noise\_y$

$$\begin{pmatrix} x_1 \\ x_2 \\ \vdots \\ x_n \end{pmatrix} \sim \mathcal{N} \left[ \begin{pmatrix} 0 \\ 0 \\ \vdots \\ 0 \end{pmatrix}, \quad cov\_mat\_x \right]$$

$$\begin{pmatrix} y_1 \\ y_2 \\ \vdots \\ y_n \end{pmatrix} \sim \mathcal{N} \left[ \begin{pmatrix} 0 \\ 0 \\ \vdots \\ 0 \end{pmatrix}, \quad cov\_mat\_y \right]$$

**end function**

Here, the generative function specifies a joint probability distribution over the kernel functions, noise parameters, and output values (point locations) for both the x and y dimensions. The input is a 3-dimensional vector of previous timepoints, and x and y locations of points at those timepoints. For each output dimension, the kernel is sampled from the PCFG as described in the main text. The ComputeCovarianceMatrix call takes the sampled kernel function (which specifies the covariance between a pair of input points), and the input, and computes a matrix of the covariance between every pair of input points. The diagonal of this matrix is the variance at a particular point, and the sampled noise value is added to the diagonal. Output values are then sampled from a multivariate normal distribution parameterized by $\vec{\mu} = 0$ and the computed covariance matrix.

# 2 Supplementary Results

**Amortized inference with LSTM**

One challenge for program-learning as a theory of human inference is that it is, in general, quite computationally expensive. For instance, the LoT model requires many thousands of samples to learn many of the patterns tested in the paper. However, participants generally took only a couple seconds to make predictions in our task. One natural question, therefore, is whether inference can be amortized in, e.g., a recurrent neural network and whether this might actually provide a good account of human behavior. To test this, we trained LSTMs [3] to predict spatiotemporal sequences under two different training regimes and with varying amounts of data. The networks had 3 inputs ($x$, $y$, $t$), 3 hidden LSTM layers with 48, 32, and 24 units, and 3 output units predicting the subsequent $x$ and $y$ values at time $t + 1$ and a standard deviation term $\sigma$ representing the amount of uncertainty in its prediction. The networks were trained to maximize the log likelihood of their predicted $x$ and $y$ values at time $t + 1$, assuming that observations were drawn from normal distributions with its predicted standard deviation $\sigma$.

We trained LSTMs on either samples from the LoT model or on parametric function samples; in both cases they learned to predict sequences of length 20, as in the experiment. For the LoT model, we sampled programs and parameters from the prior. The parametric functions had the form,

$$x_{t+1} = \alpha_{x,0} + \beta_{x,1}(t - \alpha_{x,1}) + \beta_{x,2}(x_t - \alpha_{x,2}) + \beta_{x,3}(y_t - \alpha_{x,3})$$

$$y_{t+1} = \alpha_{y,0} + \beta_{y,1}(t - \alpha_{y,1}) + \beta_{y,2}(x_t - \alpha_{y,2}) + \beta_{y,3}(y_t - \alpha_{y,3})$$

where the $\beta$ coefficients are regressors on the previous $x$, $y$, and $t$ values and the $\alpha$ parameters are offsets. The $\beta$ coefficients were each drawn from a $\mathbb{N}(0, 1)$ and the $\alpha$ coefficients were drawn from $\mathbb{N}(0, 10)$. The $x$ and $y$ values were then standardized. Each model was trained with 1,000, 4,000, and

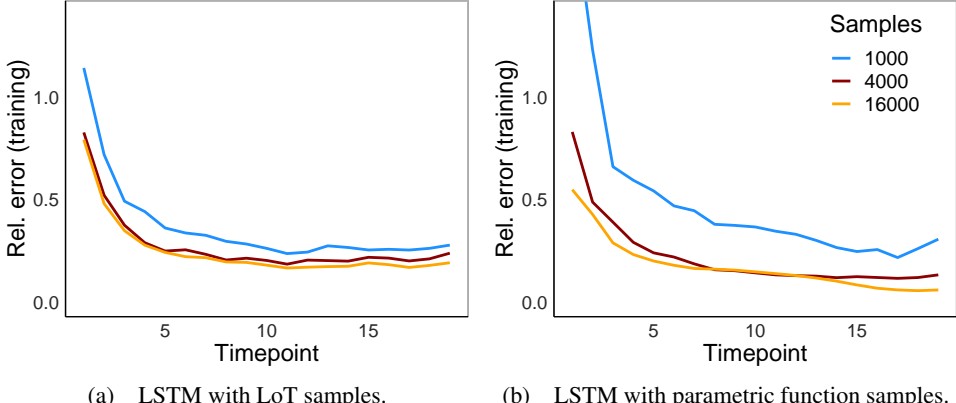

(a)    LSTM with LoT samples.    (b)    LSTM with parametric function samples.

Figure S4: Relative error in predicting trianing set examples for (a) the LSTM trained on LoT model samples and (b) the LSTM trained on parametric function samples.

16,000 examples (with 20 time-steps per example), with a 9:1 train/validation split. Each model was trained for a maximum of 2,000 epochs with an early stopping rule applied to validation set accuracy.

The LSTMs learned to predict the training data fairly well, as evidenced by the learning curves shown in Fig. S4, which show their mean relative error in predicting subsequent $x/y$ values over the course of each training sequence. There was a slight benefit of additional training examples in both cases. The mean relative error across timepoints of the LSTM trained on LoT samples was 0.37, 0.27, and 0.25 across all examples for training sets of 1,000, 4,000, and 16,000 samples respectively. The LSTM trained on parametric function samples had relative errors of 0.5, 0.22 and 0.18.

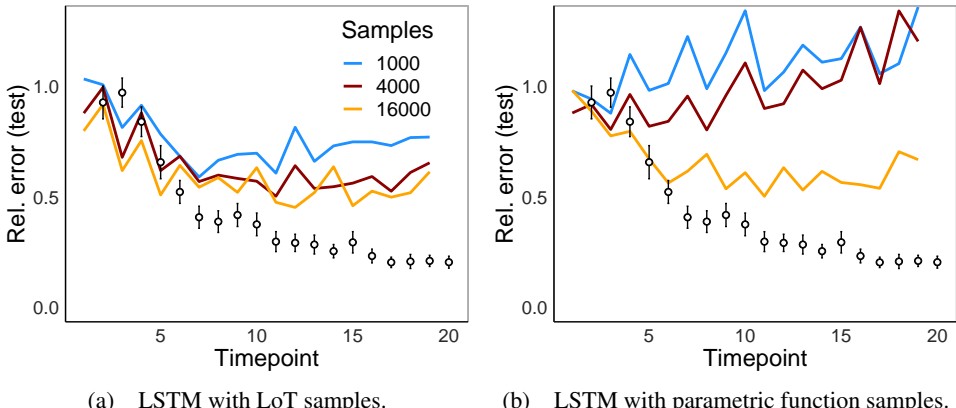

(a)    LSTM with LoT samples.    (b)    LSTM with parametric function samples.

Figure S5: Relative error in predicting examples from the experiment for (a) the LSTM trained on LoT model samples and (b) the LSTM trained on parametric function samples.

The models learned to predict the stimuli used in the experiment to some degree as well, though not as well as the models considered in the main text, nor as well as people. Figure S5 shows their mean relative error in prediction over the course of each sequence used in the experiment, clearly showing some learning over time for the LSTM trained on LoT samples and for the parametric model when trained with 16,000 examples. The LSTM trained on LoT samples had a mean relative error of 0.76 0.64, and 0.59 across all experimental stimuli when trained on 1,000, 4,000, and 16,000 examples. The LSTM trained on parametric function samples had a relative errors of 1.07, 0.96 and 0.65.

**Model fits**

Table S1 shows the results of all models fit to the data, including the inferred parameters (motor noise $\sigma_m$, attentional lapse rate $\theta_l$, lapse noise $\sigma_l$), the log likelihood of the data, and the $\Delta AIC$ from the best-fitting model (the LoT model). All models described in the main text fit better than all the additional models only reported here. While it may be surprising that, e.g., the higher-degree polynomial Ridge models fit worse than the linear-only model, they had strange biases toward curvature that were not latent in the human data. So even though the third-degree polynomial Ridge regression could fit most sequences by the final datapoint, it did not show human-like biases and error patterns.

| Model type | $\sigma_m$ | $\theta_l$ | $\sigma_l$ | LL | $\Delta$AIC |
|---|---|---|---|---|---|
| LSTM (Parametric) | 0.55 | 0.52 | 1.50 | -47868 | 52066 |
| LSTM (LoT) | 0.64 | 0.58 | 0.92 | -44841 | 46012 |
| Ridge (Deg. 3) | 0.12 | 0.45 | 0.86 | -49412 | 42272 |
| Ridge (Deg. 2) | 0.03 | 0.25 | 0.76 | -33511 | 23352 |
| GP-NC | 0.05 | 0.32 | 2.28 | -31592 | 19514 |
| Ridge (Deg. 1) | 0.01 | 0.20 | 1.96 | -30187 | 16704 |
| GP-SL | 0.16 | 0.20 | 1.60 | -28416 | 13162 |
| LoT | 0.01 | 0.24 | 1.74 | -21835 | 0 |

Table S1: Model parameters, Log Likelihood, and $\Delta$AIC from the worst-fitting model (top) to the best (bottom).

**Model predictions**

Below, we show for each model all predictions for each sequence, as well as the relative absolute error of these predictions. Videos showing human and model predictions side by side for each sequence can be accessed at `https://github.com/temills/spatiotemporal_patterns`..

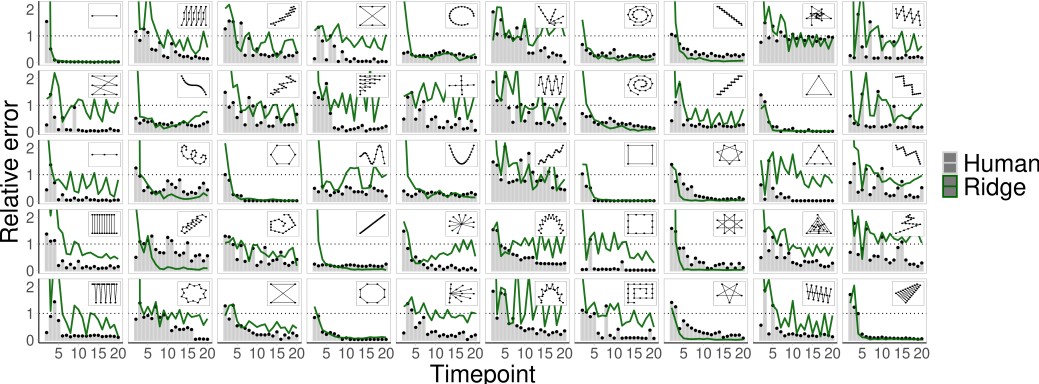

Figure S6: Absolute error relative to the distance from the previous point across all sequences tested for the **Ridge** model (green) and human data (gray bars). The sequence corresponding to the data is shown in the top-right of each panel.

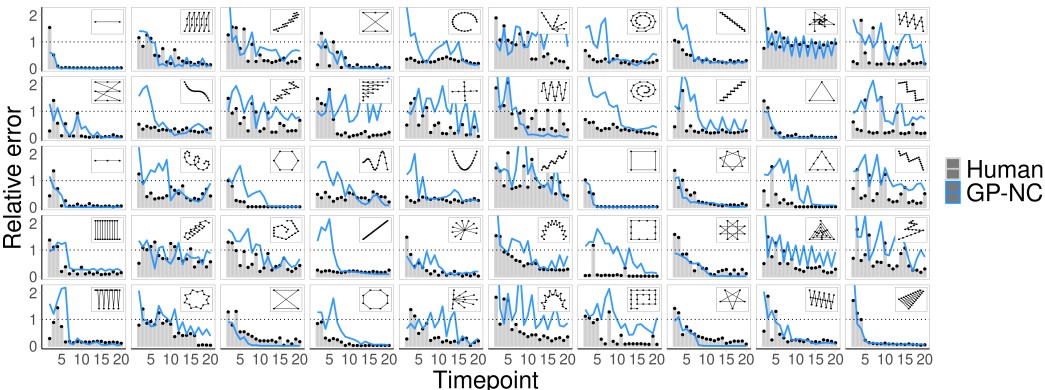

Figure S7: Absolute error relative to the distance from the previous point across all sequences tested for the **GP-NC** model (blue) and human data (gray bars). The sequence corresponding to the data is shown in the top-right of each panel.

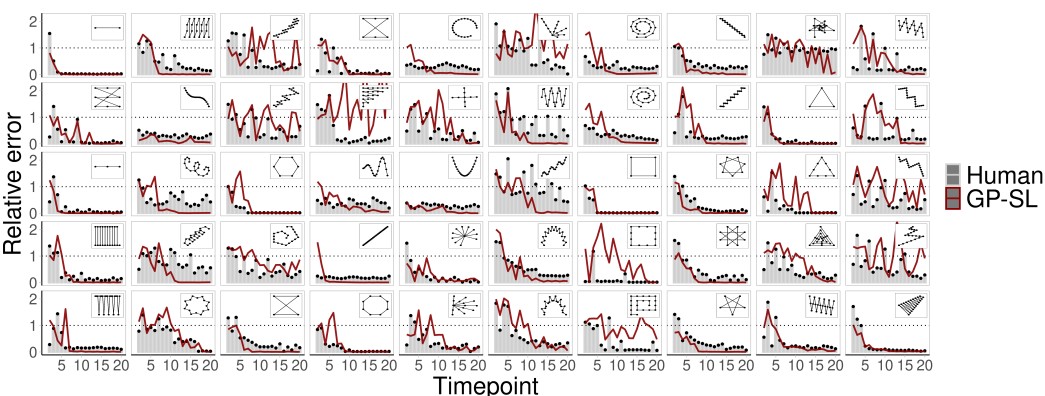

Figure S8: Absolute error relative to the distance from the previous point across all sequences tested for the **GP-SL** model (red) and human data (gray bars). The sequence corresponding to the data is shown in the top-right of each panel.

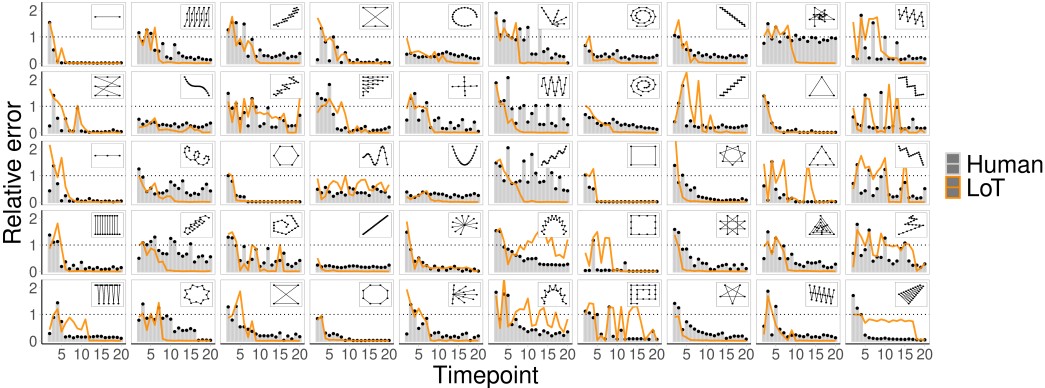

Figure S9: Absolute error relative to the distance from the previous point across all sequences tested for the **LoT** model (orange) and human data (gray bars). The sequence corresponding to the data is shown in the top-right of each panel.

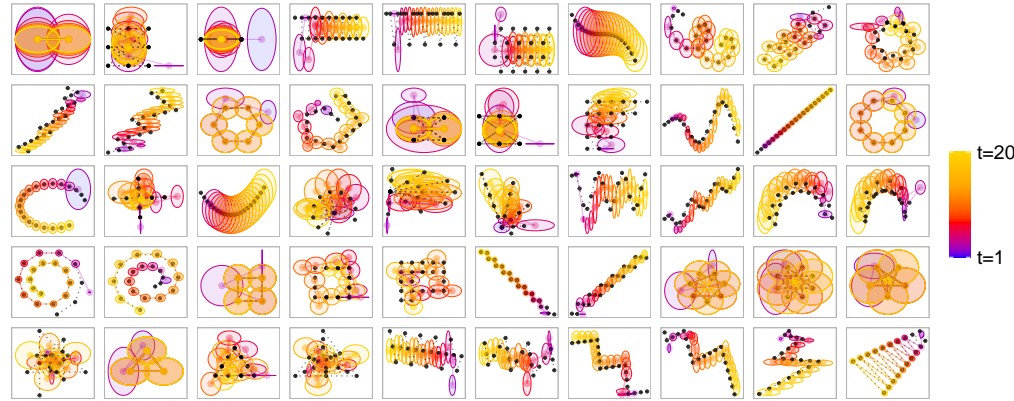

Figure S10: All predictions from the **Ridge** model for all sequences. Earlier points are cooler colors, later points are warmer colors. The size of the circles indicates uncertainty ($\pm1\sigma$.)

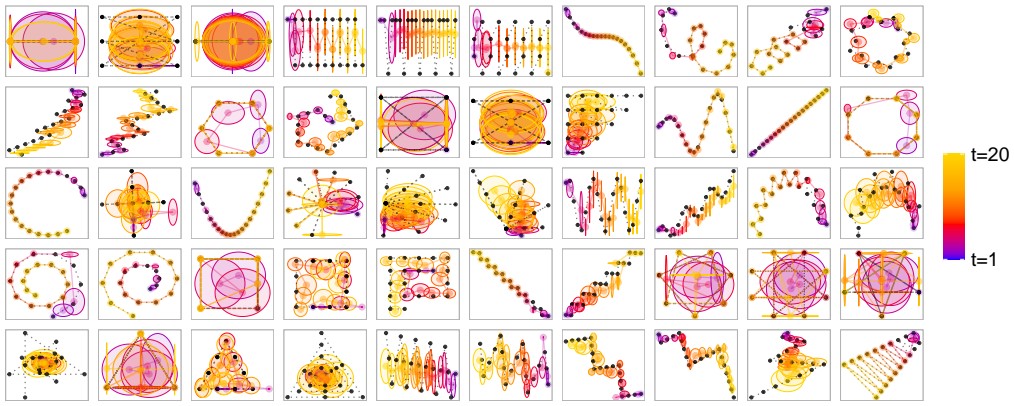

Figure S11: All predictions from the **GP-NC** model, for all sequences. Earlier points are cooler colors, later points are warmer colors. The size of the circles indicates uncertainty ($\pm1\sigma$.)

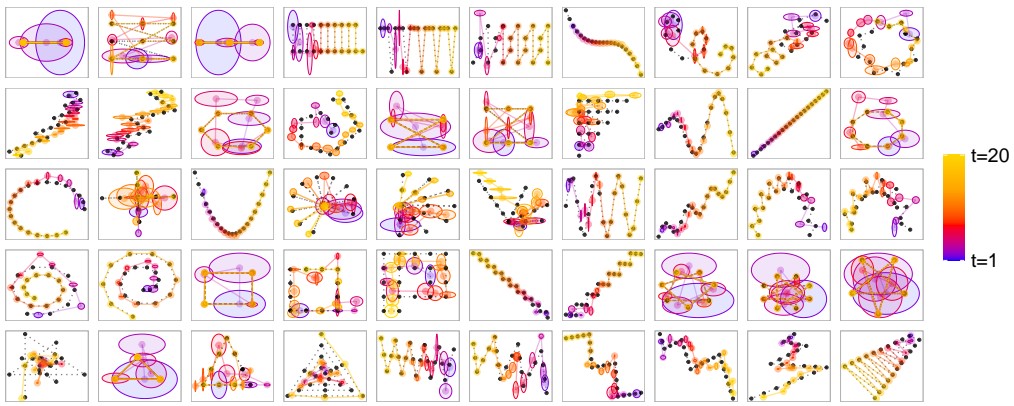

Figure S12: All predictions from the **GP-SL** model for all sequences. Earlier points are cooler colors, later points are warmer colors. The size of the circles indicates uncertainty ($\pm1\sigma$.)

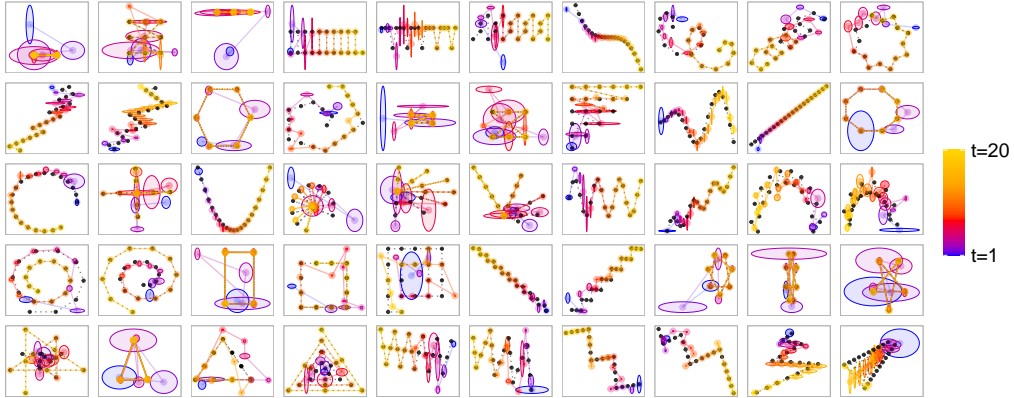

Figure S13: All predictions from the **LoT** model for all sequences. Earlier points are cooler colors, later points are warmer colors. The size of the circles indicates uncertainty ($\pm 1\sigma$.)