# OpenReview forum: "Human spatiotemporal pattern learning as probabilistic program synthesis"
_NeurIPS.cc/2023/Conference — NeurIPS 2023 poster_

### Official Review · Reviewer_zTCY · 2023-06-29

**Soundness:** 3 good
**Presentation:** 2 fair
**Contribution:** 2 fair
**Rating:** 5
**Confidence:** 2

**Summary:**

The paper presents a "Language of Thought" (LoT) model which is meant to represent how humans learn concepts from small amounts of data. To test whether the LoT model is a suitable model for human cognition the authors present a task of 2-dimensional pattern learning and collect data from human participants which are trying to solve the task. Compared to baseline models (in this case Bayesian ridge regression, a Gaussian Process with a fixed kernel structure and a Gaussian Process with learned kernel structure and hyperparameters) the LoT model seems to model more closely how humans learn sequences.

**Strengths:**

Understanding how humans can learn from small amounts of data is clearly of interest to both the machine learning community and the cognitive science community so the paper is working on an important problem.
In the limited experiments, the proposed LoT model seems to be a good fit to the data, although I have significant concerns about the empirical evaluation (see next section).

**Weaknesses:**

The four weaknesses listed below are the main reason why I recommend rejection at this point in time. Of course, I remain open to being convinced by the authors and other reviewers in the discussion period that these points do not outweigh the reasons for acceptance (EDIT: After the discussion period I still believe these are issues with the paper but the author response and the other reviews convinced me that the weaknesses don't outweigh the strengths) .

1. A large part of the value proposition of this paper hinges on the point that the proposed task of 2-dimensional pattern learning is a useful task to investigate how humans learn. However, I believe the paper needs to make it more clear why this task is suitable. Right now the setup of this task seems rather arbitrary.
2. Lacking experimental evaluation
  - There is a lack of strong/suitable baselines that are to be considered. Most of the considered baseline models are too simplistic and do not seem suitable for this task at all. I think very few people would argue that Bayesian ridge regression or a GP with a fixed kernel structure is a suitable model for human cognition. Also all the models under consideration are Bayesian models and it would be interesting to maybe see a non-Bayesian model as well. For example, one could generate a large corpus of synthetic sequences and pre-train a neural network with a suitable architecture on it. This trained neural network can then be used to make predictions.
  - At present, it is not clear which parts of the LoT model are crucial for performance. An ablation study would be interesting, for example, removing some of the primitives in the grammar and observing the effect that this has on the results. Changing the inference algorithm would be interesting as well to see how sensitive the results are to the choice of inference algorithm.
  - There are no errorbars in the experimental evaluation (especially Fig. 4a,4b). If I understand correctly in the program synthesis models the inference is performed using an approximate inference algorithm so the outputs of the inference algorithm are not deterministic.
3. I found throughout the paper that the methods are not described in enough detail. For example, there is only a very high-level, one paragraph description of the inference procedure that is used (Section titled "Program synthesis models"). Further, it would be good if there would be a more formal description of the grammar and semantics of the program synthesis models.

**Questions:**

- The title on OpenReview and in the PDF do not match up. Which one is the correct one? There is also a typo in the title in the PDF "probabalistic".
- Line 94-95: Why did you decide to exclude these 10% of participants? Given that you only have 50 participants surely you want to use as much data as possible?

Minor:
- Second level headers hould be numbered according to the style file (e.g. Line 68 should be "2.1 Method")
- Line 75-77: All the chosen numbers seem really arbitrary. How did you settle on exactly this parameter setting?
- In Figure 3 it would be good to have an x-axis. Maybe provide an ID for each sequence (what ID corresponds to which sequence can be shown in the Appendix and doesn't have to be in the main paper).
- Line 183: In "Sable-Meyer et al. (2022)" you are using a different reference than in the rest of the paper.
- Line 232: This is the correct way to do sub-headings but the numbering is off because the rest of the sub-headings are not numbered.

**Limitations:**

The paper mainly argues for the hypothesis that human learning is similar to "Bayesian program learning" and it cites a wide number of supporting papers that argue for this hypothesis. However, there is limited discussion of related work which considers alternative views and the paper would be strengthened by mentioning how the Bayesian program learning approach fits into the general wider field of cognitive science.

On the context of related work, how exactly does the proposed LoT model differ from Sable-Meyer et al. (2022)? Since the LoT seems to be the main innovation of this work it is important to clearly state how it is different from prior art. I think it is necessary to provide more details than just stating "Our LoT grammar is organized similarly, though has some added complexity [...]" (Line 187).

---

> ### Author Rebuttal · Authors · 2023-08-09
>
> *Weaknesses*
> - As to the suitability of the task, we have expanded our introduction and discussion to more clearly describe the general context discussed in our meta-response, as well as the work of Amalric et al. (2017) who looked at how children learn simple patterns on an octagon, and that of Sablé-Meyer et al. (2021, 2022), who examined how people and non-human primates perceive structure in geometric shapes, which our task extends in a few ways. The domain used in the Amalric (2017) paper is extremely simple (1-D) and only allows a very small hypothesis space, so the inference problem is fairly trivial. Similarly, Sablé-Meyer (2022) used an outlier-detection task to study what shapes people find simple and complex, which does not address the problem of inference. Our LoT model is also far less restricted in the forms it can generate, and allows for uncertainty over possible hypotheses and probabilistic outputs. Our task is also interesting because predicting where something will go is extremely simple to understand (we are currently testing young children and non-human primates on this task). However, it is also a very non-trivial learning problem; the problem of function learning from little data is very much unsolved and an active area of research in machine learning (e.g. Saad et al., 2021).
> - As we have now clarified in the paper, we chose the baseline models because they have been considered by previous researchers studying human function learning from small amounts of data (e.g., Griffiths et al., 2008; Lucas et al., 2015; Li et al., 2023),  and are common tools for data scientists applied to a wide variety of problems where there is not enough data to train a neural network. We address training a neural network in our meta-response.
> - We are currently pursuing two directions for analyzing which parts of the LoT model are most crucial for emulating people’s inductive biases. First, to specifically test the relative importance of each primitive in the current grammar, we are fitting the prior over the grammar to participant data and plan to include this analysis in the final version of the paper. Second, we are comparing different inference parameters. Given the right parameter settings (enough SMC particles or proposal steps) the choice of inference algorithm should not affect the results. Here, we use SMC with MCMC rejuvenation moves, and plan to additionally update the paper to include an analysis of how variations in the number of MCMC rejuvenation moves and SMC particles impacts our results.
> - The errorbars on human data in Figures 4a/b correspond to uncertainty about the true population mean accuracy. Although the models’ particles approximate the true posterior, variation in the predictions of those particles provides information about the distribution of the estimated posterior predictive, rather than uncertainty about its accuracy. Errorbars on the model and human data would therefore express different things and may be confusing.
> - To describe the methods in more detail despite space constraints, we have slightly expanded our explanation of the program synthesis models in the main text, though our supplemental materials section provides a more in-depth and technical description of the methods we used for all the models.
>
> *Questions*
> - We excluded 10% of participants because when using online platforms, it is assumed that some non-zero percentage of participants will provide low quality data due to lack of attention or engagement. Studies examining online workers’ attention have found that 10-20% of Prolific workers fail attention checks (e.g. Peer et. al, 2022) and we found a similar rate of seemingly random responses in a pilot. This particular experiment made attention checks difficult, so we simply excluded a sample of participants that we hoped would remove the low effort individuals while preserving most of the rest. We accounted for this in our sample size, and have on average 18 unique human predictions per timestep per sequence after excluding participants.
> - We chose our experimental parameters such that the distances between all pairs of successive points were no more than the maximum and no less than the minimum allowed distances between a participant’s guess and the previous point, and the correctness window was chosen to be as small as possible without motor errors impeding success. Importantly our model comparisons do not rely on whether participant’s first guesses were deemed correct but merely on their actual locations, and we additionally fit a motor error term for each model.
> - We have updated Figure 3 to include sequence IDs as suggested.
>
> *Limitations*
> - As to our discussion of related work, we have sought to situate Bayesian program learning within the wider AI and cognitive science literatures in our introduction (how aspects of human learning present a challenge for certain standard classes of machine learning on lines 17-34, how Bayesian learning in general has addressed some of these challenges in function learning on lines 40-46, and why Bayesian program learning is especially promising for our task given past work on geometric shape perception on lines 47-56). We have also added discussion of classical rule-based approaches to concept learning and function learning in particular (e.g. Brehmer et al., 1974), prototype/exemplar-views (e.g. Nosofsky, 1992), as well as data driven approaches like the connectionism of the 1980s/1990s (e.g. Rumelhart, McClelland, & Hinton, 1986) and more recent neural network approaches (e.g. Battelday, Peterson, & Griffiths, 2020).
> - We have more specifically stated the differences between the grammar of the LOT model and that of Sable-Meyer et al. (2022). Line 187 now reads: “Our LoT grammar is organized similarly, though has some added complexity (e.g. access to inner-state variables such as the current heading or timestep) to allow for greater flexibility in the range of sequences it can generate.”

---

> > ### Comment · Reviewer_zTCY · 2023-08-15
> > **Response to Rebuttal**
> >
> > I would like to thank the authors for their detailed response to my review and also to the other reviews. I have carefully read through the other reviews and the corresponding responses. My initial low rating of the paper was based on the lack of clear motivation for choosing the task at hand and the lack of suitable baselines. However, my low confidence score was meant to reflect that I am not an expert in the cognitive science literature. While I see that the other reviewers have similar concerns about the motivation and lack of empirical evaluation, they do weigh those concerns less heavily than I do. Furthermore, I appreciate that in your rebuttal you provide more context for why this particular task and the baseline were chosen. Consequently, I have now preliminarily improved my rating for this paper.
> >
> > I would heavily encourage the authors to include the results of the additional experiments they are conducting (i.e. the LSTM model mentioned in the general response and the ablations of the LoT model) in the final camera-ready version of this paper. Due to lack of space these results might have to be presented in the Appendix but I believe they would be very informative.
> >
> > > variation in the predictions of those particles provides information about the distribution of the estimated posterior predictive, rather than uncertainty about its accuracy.
> >
> > Apologies for being unclear in my original review about what types of errorbars I am referring to. I was not proposing to plot the standard deviation of the particles of the inference algorithm. Rather, my point was that the inference algorithm is not deterministic so it is important to have several independent runs with different random seeds to get an understanding for how robust the results are. From these repeated runs it is then possible to calculate statistics such as the standard errors or confidence intervals.

---

> > > ### Author Response · Authors · 2023-08-21
> > >
> > > > I would like to thank the authors for their detailed response to my review and also to the other reviews. I have carefully read through the other reviews and the corresponding responses. My initial low rating of the paper was based on the lack of clear motivation for choosing the task at hand and the lack of suitable baselines. However, my low confidence score was meant to reflect that I am not an expert in the cognitive science literature. While I see that the other reviewers have similar concerns about the motivation and lack of empirical evaluation, they do weigh those concerns less heavily than I do. Furthermore, I appreciate that in your rebuttal you provide more context for why this particular task and the baseline were chosen. Consequently, I have now preliminarily improved my rating for this paper.
> > >
> > > Thank you, we appreciate that!
> > >
> > > > I would heavily encourage the authors to include the results of the additional experiments they are conducting (i.e. the LSTM model mentioned in the general response and the ablations of the LoT model) in the final camera-ready version of this paper. Due to lack of space these results might have to be presented in the Appendix but I believe they would be very informative.
> > >
> > > Yes, we are planning on this.
> > >
> > > > Apologies for being unclear in my original review about what types of errorbars I am referring to. I was not proposing to plot the standard deviation of the particles of the inference algorithm. Rather, my point was that the inference algorithm is not deterministic so it is important to have several independent runs with different random seeds to get an understanding for how robust the results are. From these repeated runs it is then possible to calculate statistics such as the standard errors or confidence intervals.
> > >
> > > Okay, thanks for clarifying, this makes sense. We are in the process of running the model with multiple seeds to analyze the variance, as you suggest. We will include this analysis in the final version.

---

### Official Review · Reviewer_y3JJ · 2023-07-03

**Soundness:** 3 good
**Presentation:** 2 fair
**Contribution:** 2 fair
**Rating:** 6
**Confidence:** 4

**Summary:**

This work investigates a visual sequence prediction task in humans where a dot would move on a screen based on some kind of underlying function and the participant must predict where the dot will go next. The goal of the study is to understand people's underlying inductive biases, so only people's first attempts at the predictions for each sequences are used. Once they collected a dataset from an online experiment, they used different models for the behavioral data based Bayesian Ridge Regression, Gaussian Process Regression (with different kernels), and an "LoT" model that includes program induction primitives that implement motor drawing programs. They found the LoT model fits the human data the best, supporting a popular hypothesis in the literature that people use a"program-like" representation to represent complex structure in novel environments.

**Strengths:**

* The paper is very clearly written and it was easy to follow.

* The models seemed to be solidly implemented to explain the human behavior and the evaluation seems rigorous as well.

* Task is interesting and dataset seems to be properly collected.

**Weaknesses:**

* I think the biggest weakness of this work is the choice of patterns used in the task. It's not clear to me how exactly they were chosen (I wasn't able to find any details on how the sequence stimuli were chosen). From what I see in Fig. S3, it seems to be the kind of stimuli that the LoT model can represent well (lots of repeated geometric motifs). Given this, it is not that surprising to me that the LoT model captured human behavior well, because the underlying stimuli are ones that fit well with the model in the first place. It would be more compelling to me if the setup was something more like: use each of the models (GPs with different kernels, LoT, Bayesian ridge, etc) to somehow *generate* the sequence stimuli and *then* see that the set of stimuli the LoT model generates is the one easiest for humans to perform.

* I do disagree with the authors that because the LoT predicts behavior the best, then people's underlying representations/algorithms are program-like. It may be true that the LoT model has enough representational flexibility to capture/explain human behavior (or their *inductive bias*) well, but that does not necessarily mean people have programs in their mind that they're using to perform the task. I think it's totally plausible that you can, for example, train an RNN on a sequence prediction task on a bunch of stimuli like in Fig S3 and best predict its behavior using LoT model, without the RNN having an explicit representation of programs. To make a representational claim, I think that would require an analysis of RTs, targeted psychophysical manipulations of the underlying stimuli, or neural data. I do think explaining the high-level behavior/inductive bias is important in its own right, so I encourage the authors to redirect their claims to center more around people's inductive biases rather than their underlying representations/algorithms. That is still a valuable contribution in its own right.

* I think the figures could use a bit more clarity. I like how the authors put the original ground truth sequences as in inset in the plots in Figure 5. It'd be really helpful to see this in Figs 2, 3, and 6 as well. Otherwise it's hard to put people's predictions in the correct context without knowing what they're trying to predict in the first place. Also I would really like if Figure 2 was much bigger. It's hard to see all the nuances of the predictions. Maybe the authors can use the extra page given in the rebuttal period to make it bigger?



**Questions:**

* My biggest question is how the stimuli were chosen? There should be a principled reason for the choice of stimuli. I'm worried that only choosing stimuli with geometric/recursive structure produces a potential confound in the findings.

**Limitations:**

* The authors do well in noting the limitation of their models in having a pre-specified hypothesis space.

* I do encourage the authors to note that predicting behavior is more of a study on high-level behavior/inductive bias, rather than on representation/algorithms. Representational claims will require more experimental tools like I mentioned above.

* I didn't see any broader societal impacts section. I think there could be quick thoughts summarizing that understanding the predictability of human behavior on such spatial pattern tasks can be informative for data privacy reasons.

---

> ### Author Rebuttal · Authors · 2023-08-09
>
> *Weaknesses*
>
> - With regard to your comment about the sequences we chose, have added a paragraph to the text to better explain how we chose the stimuli. Though the exact stimuli we used were somewhat ad hoc, we chose them carefully: we wanted them to be learnable by people within 20 timepoints and we wanted variability in how difficult to learn they were (see Figure 3). As you note, we also chose them so that they could be described algorithmically using control structures, such as concatenation, loops, and recursion. While the point about the stimuli being well-suited to a LoT and therefore potentially biasing our results is reasonable, we note that it’s not actually the case that the LoT model learned them faster on average than the GP-SL model (see Figure 4). Combinations of GP kernels can learn arbitrary repeating motifs; in fact, an unbounded polynomial regression could learn all of the functions we tested also. What our experiment aimed to test, and what our results show, is that people seemed to have inductive biases that are better explained by the LoT model than the other models we tested.
> - One possibility you might be alluding to, though, is that by having participants each learn 20 sequences — rather than just a single function — their prior shifted toward functions that are easily described in a LoT, rather than (e.g.) polynomials or a combination of GP kernels. However, if this were the case we should expect that the LoT model would fit better as participants do more trials. We tested whether this was the case and found that it’s not: there’s no effect of trial number on the goodness-of-fit of the LoT model (in log likelihood and MSE), meaning if participants were adapting over the course of the experiment, it was not adaptation specifically to an inductive bias more like the LoT model we used. To further test this, however, are currently working on running a follow-up experiment using stimuli generated by sampling functions from the GP-SL model.
> - As to your concern regarding how we frame our results, we agree that our experiment and analyses are essentially aimed at understanding people's inductive biases and were not aimed at addressing the underlying algorithms or representations. We tried to make this clear in how we presented the questions we were after, but we realize that we were somewhat inconsistent about this ---specifically in our final paragraph of the introduction where we describe our experiment and modeling as testing “competing accounts of the underlying representations.” We have changed this and a couple other sentences throughout the paper to make clear that our analyses are aimed at understanding the nature of people's inductive biases, rather than the algorithms and representations supporting them.
> - Regarding the point about the figures, we also like the insets on Fig. 5 (thanks!)  but space limitations made putting them other figures tricky — they take up a fair amount of space within the facets. We will work on adding them to Figs 2, 3, and 6 though because we agree it would add clarity. Likewise, space constrains the size of Figure 2, making it difficult to extract information about how people responded to each function at each time point. However, our aim with Figure 2 was to highlight the nature of the task, the types of sequences we used as stimuli, and the change in the variability of responses early on (in blue/purple) to later in each sequence (in red/yellow) — which we think it does successfully. We added a large version of Figure 2 to the extra page, as you requested.
>
> *Questions*
>
> Regarding stimuli, see first point above.
>
>
> *Limitations*
>
>
> To your point about broader impacts, thanks for bringing this up. While it is not immediately obvious to us how our results might be applied to data privacy, we are curious to hear your thoughts on this. As to discussing other societal impacts, we consider our results to be relevant to core questions in cognitive science about how people infer rich latent structure from limited data, and thus to bear on many socially impactful domains (e.g. pedagogy).  However, given space constraints, we did not consider any single domain to be relevant enough to warrant explicit discussion.

---

### Official Review · Reviewer_XJBu · 2023-07-05

**Soundness:** 3 good
**Presentation:** 4 excellent
**Contribution:** 3 good
**Rating:** 6
**Confidence:** 4

**Summary:**

The paper investigates whether humans learn visual patterns by programming, i.e., inducing probabilistic models to fit observed data.
In a synthesized visual prediction task, they evaluate human performance against standard parametric and non-parametric time-series models and Bayesian program synthesis models. They found that human learning resembles the "Language of Thought" model, supporting the hypothesis that human learning can be understood as inference over an expressive space of programs.

**Strengths:**

The paper conducts a very interesting investigation and provides valuable insights on human pattern learning.

Their experiment method is interesting and appropriate and carefully carried out.

The paper is well written and I enjoy reading it.

**Weaknesses:**

I didn't find any major weakness.

There are some typos.
E.g., line-26 "oft-cited"

**Questions:**

The paper hasn't discussed the limitations of their experiment method and their empirical results and implications.
So I encourage them to add remarks about this aspect.

I tend to accept this paper.
I didn't give a higher score because I am not very familiar with related work of this particular area.

**Limitations:**

No ethical issue. See Questions for other limitations.

---

> ### Author Rebuttal · Authors · 2023-08-09
>
> *Strengths*
>
> - *The paper conducts a very interesting investigation and provides valuable insights on human pattern learning.*
> - *Their experiment method is interesting and appropriate and carefully carried out.*
> - *The paper is well written and I enjoy reading it.*
>
> Thanks!
>
>
> *Questions*
>
> *"The paper hasn't discussed the limitations of their experiment method and their empirical results and implications. So I encourage them to add remarks about this aspect."*
>
> One way in which our method limits the implications of our results is the fact that the patterns we tested are rather geometrically regular and free of noise, potentially making the inductive biases of the LoT model especially well-suited to the task. We bring up this point in our last paragraph beginning on line 318, but have changed the beginning of this sentence to call attention to its relevance in determining the implications of our results. This paragraph now begins: “We have shown that during 2D pattern learning, people display inductive biases favoring geometric regularity and precise algorithmic structure which are well captured by program-like representations. One limitation of the current paradigm is the fact that people’s inductive biases may be shaped during the experiment by the patterns we tested, which indeed all had some geometric regularity and were free of noise. We note, however, that the fit of participant responses under the LoT model did not improve as they completed more trials, making it unlikely that this sort of adaptation played a significant role in driving the results.”
>
> In future work, we hope to test how people’s inductive biases adapt based on task-dependent and experience-dependent factors.
>
> In this last paragraph, we also discuss an important limitation of our models beginning on line 320: "Indeed, the ability to construct and update domain-specific hypothesis spaces is a remarkable feat of human learning which the field of program synthesis has only begun to make progress toward incorporating — here, both the GP and LoT models do inference in strictly pre-specified hypothesis spaces."

---

> > ### Comment · Reviewer_XJBu · 2023-08-17
> >
> > Thank you for your response. I encourage you to add the limitation discussion into the final version.

---

### Official Review · Reviewer_igwV · 2023-07-07

**Soundness:** 3 good
**Presentation:** 4 excellent
**Contribution:** 3 good
**Rating:** 7
**Confidence:** 4

**Summary:**

The paper tests human online prediction and learning in a moving dot task against a variety of probabilistic models. The paper primarily reports its behavioral results from human subjects in Section 2, then describes probabilistic models based on ridge regression, compositional and non-compositional Gaussian processes and a "Language of Thought" hypothesis in Section 3.  The "language of thought" model provides the best model fit in terms of log marginal likelihood.

**Strengths:**

The paper manages to find a clear correlation between the relative error score by human participants and that of the language of thought model, while still including Gaussian process models used in prior function-learning experiments.  Two possible compositional models are considered rather than just one.

**Weaknesses:**

Only one "language of thought" hypothesis is given, and its design is ad-hoc and lacking in clear neural or motoric motivation, despite the paper attempting to take inspiration from computational modeling of drawing motor skills.

**Questions:**

Can the authors please use the same inference technique for the various probabilistic models?  It may seem like overkill to conduct a ridge regression by Resample-Involutive-Move MCMC, but it helps to identify which aspects of the model fit derive from the quality of the inference algorithm and which come from the model structure itself.

**Limitations:**

While this behavioral experiment and numerous others confirm the ability of the brain to learn and deploy program-like representations, the model comparison in this paper remains unfair when between a program-like representation and a generic Gaussian process or polynomial regression.  Which aspects, specifically, would the authors hypothesize are program-like in what the brain actually constructs?  How can variable binding be at work in a spatial prediction task?  Which aspects of this task are structured to elicit symbolic reasoning vs sensorimotor inference?

---

> ### Author Rebuttal · Authors · 2023-08-09
>
> *Weaknesses:*
> *“Only one 'language of thought' hypothesis is given, and its design is ad-hoc and lacking in clear neural or motoric motivation, despite the paper attempting to take inspiration from computational modeling of drawing motor skills.”*
>
> This work primarily aims to characterize people’s inductive biases during 2-D sequence learning, and though we acknowledge that motor skills affect performance on our task, we fit motor error terms for each of our models rather than modeling these motor skills themselves. We do not intend to take inspiration from models of drawing motor skills, but we realize our description of a “drawing language” employed in past work beginning on line 183 (“The grammar draws inspiration from Sable-Meyer et al. (2022), which defined a simple drawing language used to model human representations of geometric shapes’’) may have been confusing. We have added an additional sentence after this one to clarify the nature of this work (“Specifically, the complexity of MDL programs in this grammar were found to predict people’s ease of encoding and recalling the shapes generated by those programs’’).
>
> The choice of our LoT grammar was motivated by this past work. One reason we did not additionally test alternative LoT grammars is because the current grammar is quite general. It includes programmatic control structures (e.g. concatenation and repetition), arithmetic operators, and numerical values and variables. These primitives can describe general programmatic structures which define sequences of dots when combined with domain-specific action primitives, such as Turn or Move operators. However, we agree that people likely do not have exactly the inductive bias specified by this particular language (and that people may in fact vary in their inductive bias). We are currently working on fitting participants’ priors under this grammar and also comparing the fit of this model to subsets and supersets of this particular grammar — we will include these analyses in the final version.
>
>
> *Questions:*
> “Can the authors please use the same inference technique for the various probabilistic models? It may seem like overkill to conduct a ridge regression by Resample-Involutive-Move MCMC, but it helps to identify which aspects of the model fit derive from the quality of the inference algorithm and which come from the model structure itself.”
>
> We agree that although using a more complex inference algorithm such as Resample-Involutive-Move MCMC should not be expected to affect the predictions of the Bayesian ridge regression or fixed-kernel gaussian process models, it is good practice to keep this constant across models. We have updated the implementation of these baseline models accordingly.
>
> *Limitations:*
> *“While this behavioral experiment and numerous others confirm the ability of the brain to learn and deploy program-like representations, the model comparison in this paper remains unfair when between a program-like representation and a generic Gaussian process or polynomial regression. Which aspects, specifically, would the authors hypothesize are program-like in what the brain actually constructs? How can variable binding be at work in a spatial prediction task? Which aspects of this task are structured to elicit symbolic reasoning vs sensorimotor inference?”*
>
> The Bayesian ridge regression and fixed-kernel gaussian process models have been considered by previous researchers studying human function learning and are common tools for data scientists applied to a wide variety of problems, making them reasonable choices as baseline models. However, we agree that they lack much of the flexibility displayed by humans and afforded by the Bayesian program learning models. We thus consider the most interesting comparison in our paper to be between the LoT and Gaussian Process program learning models.
>
> This comparison in particular supports our conclusion that people’s inductive biases are structured and algorithmic (i.e. “program-like”), as captured by the LoT model — this is primarily a computational-level claim.
>
> As to your question about variable binding, the fact that people can learn patterns of the form “Repeat X,” where X is some arbitrary motif, seems like a straightforward example of people’s ability to assign a value to a variable for spatial prediction.
>
> One aspect of our results which we find especially interesting is the evidence for “symbolic reasoning” in a “sensorimotor inference” task. While the task feels at once rather like it engages low-level, automatic processes latent in early vision, our results suggest that people are pursuing symbolic, program-like hypotheses. The question of how people might synthesize and evaluate such hypotheses online is especially interesting given the computational challenge of navigating large, discrete search spaces. Relatedly, it is unclear to what extent general visual recognition mechanisms might play a role in participants’ ability to “see” structure in patterns.

---

> > ### Comment · Reviewer_igwV · 2023-08-21
> >
> > I thank the authors for their engaging rebuttal, and especially for updating the baselines to use the same inference algorithm(s).  While I already rate the paper as an Accept, I will be raising my confidence.

---

### Official Review · Reviewer_iZWm · 2023-07-25

**Soundness:** 4 excellent
**Presentation:** 3 good
**Contribution:** 3 good
**Rating:** 7
**Confidence:** 3

**Summary:**

In this paper, the authors study human function learning as an algorithmically structured process, comparing compositional models to behavior. Their experimental protocol is a spatial variant of function learning from the psychological literature, and requires participants to learn a sequence or function by iteratively predicting the next data point, given some history of data points. The authors use two novel models to analyze behavior in this domain: a compositional Gaussian Process (GP) which involves inference over compositionally-structured kernels, and a Language of Thought (LoT) model which performs Bayesian inference over programs inspired by a language for drawing shapes. They compare these models against Bayesian regression, a simple baseline, and Gaussian Processes with fixed kernels, which is a standard approach in contemporary models of function learning (introduced to this domain in Griffiths et al. 2008).

In the experiment, participants are effective at learning the presented functions. Model comparison to behavior resulted in the LoT model as most predictive of behavior, followed by compositional GP, then either fixed-kernel GP or Bayesian regression. Models were compared to behavior in terms of overall accuracy on the tasks, as well as choice models that fit noise terms.

**Strengths:**

The authors study human function learning by introducing and/or developing two novel computational models. Their results are quantitatively compelling and also include helpful qualitative demonstrations relating their models to behavior. The authors make a compelling case that structured hypothesis testing is an important part of how people approach this type of problem.

The models they present are sensible, but computationally novel, approaches for modeling this domain. So, I feel like the introduction / elaboration of these models is a novel contribution. In addition, this paper is novel by comparing expressive classes of models (compositional GPs and LoT) that are increasingly important tools for modeling behavior. This paper also makes a novel contribution by demonstrating the importance of structured representations in understanding human behavior.

The research methodology seems sound. It has a carefully designed experiment inspired by prior research, sound behavioral analysis (appropriate inclusion of noise terms) with simple, but standard, forms of model comparison. The modeling seems appropriately careful, with correctness checks enabled for more complex kinds of inference.

**Weaknesses:**

Broadly, my largest concerns with the paper are that connections to prior work are often left implicit. I feel like the paper makes a number of novel and interesting contributions, but it would be stronger if these contributions were better contextualized.

The task in this paper isn't explicitly related to other existing tasks in the psychological literature. Clearly describing the changes to the experiment, and their motivation, would help clarify the presentation of the experiment. To my knowledge, a novel difference from existing experiments is that the functions are 2D, or spatial, a departure from one class of prior studies that involve 1D function learning (Griffiths et al. 2008, Lucas et al. 2015, Schulz et al. 2017, Yi et al. 2023). This experiment also departs from the experiment in Amalric et al. 2017, which was spatially embedding, but restricted to functions over a small number of points.

The novel models tested in this paper (compositional GP, LoT), both seem to be drawn from the prior literature. The LoT model is explicitly noted to be drawn from a prior over geometric shapes, but differences in the grammar are only ambiguously noted ("Our LoT grammar is organized similarly, though has some added complexity to allow for greater flexibility in the range of sequences it can generate." Lines 187-188). The compositional GP model is also drawn from prior research in data modeling (Saad et al. 2019), but is also related to the model in Schulz et al. 2017 used to study function learning, which isn't explicitly noted in this section. While the Schulz et al. 2017 model is much simpler because it bounds the size of expressions, so it tests a finite number of GPs, the grammar they outline seems to be a subset of the model presented in this paper. I think there are a number of novel contributions in the modeling (LoT for function learning, inference for compositional GP, comparison of these two expressive classes of models), but the current presentation could make these contributions clearer through more explicit language comparing to prior work.

As an aside to my above comments, I recognize that the level of contextualization I am advocating for might be more appropriate for a psychological audience. However, given the extensive prior work, and also the exciting contributions in this paper, I think that computational audiences, in addition to psychological audiences, would benefit from having this additional context.



**Questions:**

- The second paragraph of the paper references a wide variety of inductive biases people, but not machines, are thought to have. How are these example biases related to the kind of learning people employ and bring in this task? I felt a bit confused to see a paragraph of reference to causal and physical inductive biases that was subsequently absent from the remainder of the paper. Instead, the discussion seems to be more focused on the role of structured representations in visual processing.
- Two standards for accuracy are used; either within 0.026 * w pixels of correct (for participants) or relative absolute error < .25 (for models). Is it possible to relate the two explicitly in the text?
- Participant mean relative absolute error is reported as < 1. Is it possible to report the actual mean, instead of a bound?
- Can the authors remark on the relatively small number of particles used in SMC (20), compared to the large number of rejuvenation steps (100,000) used for inference?
- Figure 4c-f: Since each point corresponds to a problem, is the relative error averaged across timepoints?
- Is it possible to include examples of inferred programs for both the compositional GP / LoT models in the supplement?
- How did the compositional GP and LoT models differ in terms of their predictions? I thought Figure 6 was helpful in understanding some patterns of the LoT model. From the accuracy traces in the supplement, it seems there are qualitative differences between the two models, but examples like Figure 6 could be helpful.
- Can the authors discuss how the choices in their inference algorithms might relate to the algorithms people use? For example, one recent paper (Fränken et al. 2022) notes that different classes of proposal distributions are better as an algorithmic account of concept learning in people.

Minor:

- Title "probabalistic" -> "probabilistic"
- Inconsistent style when referring to dimensionality: "one-dimensional", "3-dimensional", "2-d", "2D"
- Some stimuli in Figure S3 seem to have a constant color. I assume this means the sequence was repeated, but can this be explicitly stated somewhere?
- The authors note the choice of proposal distributions for the LoT was motivated by the "challenge for inference as it defines an extremely large, discrete search space". I'd be curious to know more explicitly if there were motivating experiences for these choices, though I'm not sure the paper should be updated based on this.

Fränken, J.-P., Theodoropoulos, N. C., & Bramley, N. R. (2022). Algorithms of adaptation in inductive inference. Cognitive Psychology.

**Limitations:**

Some limitations the authors could acknowledge:
- Would these results generalize to a broader space of stimuli?
- How are these models related to the algorithms people employ?

---

> ### Author Rebuttal · Authors · 2023-08-09
>
> *Weaknesses*
> - Regarding contextualization and background, we agree that the current paper will be improved by more clearly contextualizing its contributions and the ways it differs from prior work. Though limited by space constraints, we have restructured our introduction and discussion to more explicitly state the contributions of past work, how our task was chosen to build on this work, and how our results should be interpreted with this context in mind.
> - We have attempted to clarify the difference between our task and prior work on 1-D function learning, as well as how it combines and builds on insights from both Amalric et al. (2017), as you mention, and Sablé-Meyer (2022). See global response for more detail.
> - Regarding comparisons to prior work, we have explained our work in the context of Sablé-Meyer (2022) in more detail in the introduction, as described in our previous response, and also updated our description of differences between the grammars on line 187 to be more specific: “Our LoT grammar is organized similarly, though has some added complexity (e.g. access to inner-state variables such as the current heading or timestep) to allow for greater flexibility in the range of sequences it can generate”.
> - We have also updated our description of the GP program learning model to explain its relation to the model in Schulz et al. 2017. Thanks for pointing this out.
>
> *Questions*
> - With regard to your question about the passage from the introduction, this paragraph was intended to highlight previous work that has examined how people and DNNs trained on visual data seem to have different visual inductive biases seemingly due to people imposing a greater degree of physically-coherent structure on observed scenes. But, as we were trying to say, people seem to impose structure on visual patterns that goes beyond physical sense-making, which is a significant motivation for our study. We modified a sentence in the discussion to highlight that people’s ability to perceive latent structure in scenes goes beyond making it physically-coherent.
> - With regard to your question about accuracy measures, we changed Figure 4b to be the proportion of predictions (for both people and models) within 0.026 * w pixels. We also added some additional descriptive statistics about both absolute and relative error to both the human and modeling results sections.
> - To your other question about error, mean relative absolute error overall is 0.43. On the very first timepoint the mean relative error is 0.94 and on the final timepoint it is 0.21.
> - To your question about particles and rejuvenation steps, it is certainly true that having both more particles and more rejuvenation steps will, in general, make inference better. In this case, though, the total number of MCMC rejuvenation steps is vastly more important to inference than the number of particles. This is because the particles are initially just samples from the prior, and given the vast search space, it is very unlikely that any sample from the prior will be close in program-space to the true generative function — so the total number of steps of MCMC to search for good hypotheses ends up essentially as the limiting factor in approximating the posterior.
> - To the question about averaging relative error, you’re right that the relative error is averaged across timepoints and we have updated the caption to make this more clear.
> - To the question about including examples of programs, yes: we have updated the supplement to include example programs for each model.
> - Regarding how the LoT and GP-SL models biases differed, this is a pretty important point that we discussed only at a high level due to space constraints. Both models are able to learn all the functions eventually, but do so at different speeds for different functions, and have different latent biases. In general, the GP-SL model has much weaker biases toward certain “simple” angles (e.g. right-angles) and exact repetition of previously observed movements.  The LoT model is more strongly biased toward lines, repeated exact changes, and simple angles (90-degrees, 45-degrees); it also learns patterns with recursive structure (e.g. functions with incrementing numbers of points before turning) more quickly.
> - To the question about inference algorithms and their relation to those humans might use, we should first note that the modeling we did was not really intended to capture human inference at an algorithmic level. People probably make much better proposals and consider many fewer hypotheses than the 20 particles x 100,000 steps that we used. However, as the paper you reference argues, some of the proposal mechanisms we implemented for the purpose of tractability may actually be useful in modeling how people constructively develop more complex hypotheses from simpler ones and make local edits to those hypotheses. Specifically, we used a slightly more complex version of the “Tree Surgery” proposals that Fränken et al. (2022) found best described the proposals that people made in their task. We have added a note on this link to the discussion.
>
> *Minor*
> - As to the choice of proposals for the LoT, this was essentially to make the problem of search for hypotheses tractable, by allowing a greater variety of paths to sample programs nearby in program-space.
>
> *Limitations*
> - To the point about limited types of stimuli, is certainly possible that our LoT will not capture how people learn certain types of patterns that we didn’t test, like physically-realistic motion paths. We have added a note about generalizability to the discussion.
> - As to the question about inference algorithms, as noted above, we added a couple sentences to the discussion about tree surgery, mentioning the Fränken et al. (2022) paper you pointed us to as well as other work using particle filters to model order effects in human inference (Dasgupta et al., 2020).

---

> > ### Comment · Reviewer_iZWm · 2023-08-15
> >
> > I want to thank the authors for an extremely thorough rebuttal, and appreciate the openness in addressing the concerns of all the reviewers. I think the proposed updates will more clearly situate this paper's contributions, particularly for those unfamiliar with this line of psychological study. I think the paper makes a number of technically-sophisticated improvements to modeling this class of behavior, and clearly noting what is built on, and what is extended, will make the paper's novelty that much clearer.

---

### Author Rebuttal · Authors · 2023-08-09

We would like to thank the reviewers for their thoughtful comments. We appreciate that they generally found our task and modeling work interesting and methodologically sound, even if there were some areas they thought could be improved. We will first try to summarize areas the reviewers agreed upon and ways we have improved our paper, if we were already able to do so, and additional analyses we are currently pursuing that will be complete by the “camera-ready” date.

1. Some of the reviewers expressed a desire for greater clarity in motivating the experiment and our approach more broadly. We agree with these reviewers that this paper would be improved by better motivating the task and methods, as the way it was written does make the design seem somewhat ad hoc — which it was not. We have revised the introduction and discussion to better situate our work in a long tradition of research aimed at understanding how people learn new concepts from small amounts of data (e.g. Shepard, 1961; Tenenbaum, 2001). Prior work has examined how people learn rule-based concepts (e.g.  Feldman, 1997), functional relationships (e.g. DeLosh, Busemeyer, & McDaniel, 1997), and sequences (e.g. Amalric et al., 2017), among many other domains. Much of this past work on concept learning has been aimed at determining the nature of people’s concepts (i.e., are they “rule-like” or “prototype-like” or “probabilistic”), or attempts to link properties of concepts to their learnability (e.g. higher boolean complexity ⇒ harder to learn).  One big gap in essentially all of this literature, however, is the problem of inference in large search spaces. We believe our task is interesting because it is extremely simple to understand on the one hand — even primitive animals can predict motion trajectories — but also an inferentially hard problem with a vast hypothesis space on the other. Furthermore, because this task only weakly constrains participants’ responses in a continuous space, it enables us to observe rich structure in people’s hypotheses to the extent that it exists.

2. One reviewer expressed concern that the baseline models we tested were not adequate; others suggested that a recurrent neural network trained on synthetic sequences might successfully perform our task. We first should note that the baseline models were chosen based on prior work: they were previously used as models of human function learning that prior researchers have argued might explain how people learn patterns from small amounts of data. We have clarified this point in the paper. However, we think the problem of instilling inductive biases into a neural network that allow it to succeed at our task is interesting and potentially informative. We are currently training an LSTM on thousands of sequences sampled from the LoT model’s prior and testing its performance on the experimental sequences. Our preliminary results suggest this is actually quite a hard problem: while it is able to learn simple functions (e.g. lines, curves), it is so far unable to generalize to many of the patterns used in the human experiment. Of course, with a large enough network and sufficient training data, it should eventually behave like the LoT model. But these results give us reason to think this may not be a very satisfying algorithmic-level account of people’s inductive biases, as people are able to do this task without much obviously relevant experience, while it requires significant amounts of data to train a neural network to behave like a program-learning model in this domain.

3. Relatedly, some of the reviewers had questions about the LoT model, with some seeing it as somewhat ad hoc or unmotivated, and others wondering how the grammar differed from that used in Sablé-Meyer et al. (2022). We realize we did not explain the basis for our choice of LoT model in enough detail. This model was inspired by one presented in Sablé-Meyer et al. for generating geometrical patterns, which they found predicted how easy or hard a geometrical form was for people to encode and recall. However, that model was severely restricted in the forms it could generate — e.g., it cannot produce a right triangle — and did not allow for uncertainty over possible hypotheses or probabilistic outputs. So, in brief, while our grammar is quite similar to theirs, it has been extended to allow essentially any possible pattern to be expressed. We have revised the main text (both the discussion and modeling sections) to better describe the origins of this grammar, noting both why it is a reasonable choice — with a very similar grammar having successfully modeled human geometry perception — and how it is novel.

4. Some of the reviewers expressed concern over the choice of stimuli used in our experimental task. Specifically, there was concern that the choice of “algorithmic” patterns may have biased our results in favor of the LoT model. This is a reasonable critique of the experiment. However, even if people's inductive biases were adapted to the domain, it is interesting that people’s behavior seems to reflect discrete, program-like hypotheses, as the problem of searching over a large space of discrete hypotheses is computationally difficult. Furthermore, we have good reason to think that the stimuli did not, in fact, bias the results in this way: even on the very first pattern that participants completed, participants’ responses were best fit by the LoT model — that is, participants did not seem to adapt their priors to conform to the LoT model as they completed the experiment. We realize this may only be partially convincing, though, and are currently working on an experiment with stimuli generated by sampling functions from GP-SL model.

5. We have also fixed the typos that were pointed out, as well as stylistic issues with headers and citations.

These points represent, broadly, the reviewers’ shared concerns. We have additionally addressed reviewers’ individual concerns point-by-point.

---

### Decision · Program_Chairs · 2023-09-21

**Decision:**

Accept (poster)

**Comment:**

All reviewers agree that this paper makes a solid contribution worthy of publication, reviewers engaged extensively in the rebuttal period and their opinions seem well reasoned and supported by a careful evaluation of the work.